



# Unstructured global to coastal wave modeling for the Energy Exascale Earth System Model using WAVEWATCHIII version 6.07

Steven R. Brus[1], Phillip J. Wolfram[2], Luke P. Van Roekel[1], and Jessica D. Meixner[3]

[1]Fluid Dynamics and Solid Mechanics, Los Alamos National Laboratory, Los Alamos, NM, USA
[2]Advanced Engineering Analysis, Los Alamos National Laboratory, Los Alamos, NM, USA
[3]National Centers for Environmental Prediction, Environmental Modeling Center, National Oceanic and Atmospheric Administration, College Park, MD, USA

**Correspondence:** Steven R. Brus (sbrus@lanl.gov)

**Abstract.** Wind-wave processes have generally been excluded from coupled Earth system models due to the high computational expense of spectral wave models, which resolve a frequency and direction spectrum of waves across space and time. Existing uniform-resolution wave modeling approaches used in Earth system models cannot appropriately represent wave climates from global to coastal ocean scales, largely because of tradeoffs between coastal resolution and computational costs. To resolve this challenge, we introduce a global unstructured mesh capability for the WAVEWATCHIII (WW3) model that is suitable for coupling within the U.S. Department of Energy's Energy Exascale Earth System Model (E3SM). The new unstructured WW3 global wave modeling approach can provide the accuracy of higher global resolutions in coastal areas at the relative cost of lower uniform global resolutions. This new capability enables simulation of waves at physically relevant scales as needed for coastal applications.

## 1 Introduction

Wind-generated waves play an important interfacial role in the global coupled climate system. They mediate multi-phase interactions between the ocean, atmosphere, and sea ice (Cavaleri et al., 2012) and influence the land surface in coastal zones (Mariotti and Fagherazzi, 2010). For example, wave processes drive air-sea momentum transfer (Donelan et al., 2012), enhance ocean mixing via Langmuir turbulence (Belcher et al., 2012), and modulate ocean surface albedo (Frouin et al., 2001). Ocean currents also interact with waves via Doppler shifting, which has an effect on wave heights (Ardhuin et al., 2017). Wave physics are also critical drivers of coastal processes such as wave-setup (Longuet-Higgins and Stewart, 1964), which affects coastal flooding (Dietrich et al., 2011) and sediment transport (Warner et al., 2010). In high-latitude regions, sea ice damps wave propagation, and in turn, waves fracture ice floes in the marginal ice zone (Squire et al., 1995). Waves also drive unstable currents at the ice edge, leading to mesoscale eddy generation (Dai et al., 2019).

In all of these coupled Earth system model applications, high mesh resolution is required to accurately model waves in coastal regions, largely due to the role waves play at these interfacial scales. As shown in Figure 1, wind wave periods represent a distinct portion of the energy spectrum that would be infeasible to resolve explicitly. Phase-resolving wave models (Kennedy et al., 2000), although necessary for applications such as coastal engineering, e.g., harbor design, are too expensive for use





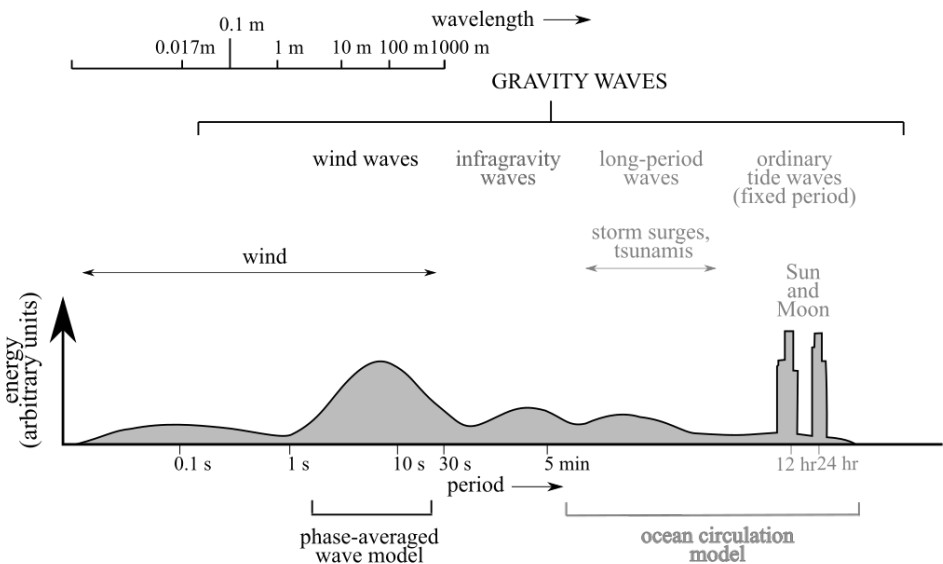

**Figure 1.** Energy spectrum for wave frequencies in the ocean showing the role of phase-averaged wave models. Adapted from Wright et al. (1999).

beyond local coastal areas. Phase-averaged approaches are applicable and appropriate for larger-scale applications because

they do not directly resolve the free surface. However, even phase-averaged wave models are quite expensive compared to the atmospheric and oceanic dynamic cores used in Earth system models.

Since phase-averaged wave models are known to be expensive, variable resolution approaches can be used to economically resolve both the coastal and global regimes. Using uniform structured meshes, the resolution will either be too coarse to accurately simulate coastal waves or the computational cost will be too great to resolve the entire global ocean. Unstructured

capabilities, however, provide the flexibility to resolve key scales relevant to accurate wave simulation in coastal regions, e.g., shallow wave breaking as a function of water depth, shoaling, refraction, and triad wave interactions (Zijlema, 2010).

Ultimately, waves must be simulated in Earth system models because the historical wave climate is evolving under decadal-scale climate changes. The global wave climate has become increasingly intense as driven by increased wind speeds (Young et al., 2011) and warming sea surface temperatures (Reguero et al., 2019). This trend is expected to increase under future

greenhouse gas emission scenarios (Hemer et al., 2013; Amores and Marcos, 2020). This has important implications for assessing the impacts of climate change in coastal regions. The combination of sea level rise and more intense storms will make coastal inundation from storm surge and waves a bigger risk to coastal communities (Vousdoukas et al., 2018). A more extreme wave climate will also continue to drive coastline changes (Mentaschi et al., 2018). Furthermore, waves are also a potential source of renewable energy. Modeling the future wave climate is important to designing effective wave energy

conversion strategies (Wu et al., 2020).





Increased wave climate intensity also has consequences for high-latitude regions. Longer fetch lengths due to decreasing sea ice extent in the Arctic produce larger long period waves that are more effective at fracturing sea-ice. This has the potential to cause a feedback in which waves accelerate sea ice retreat, leading to yet longer fetch lengths (Thomson and Rogers, 2014). The loss of sea ice also increases the exposure of Antarctic ice shelves to swell waves, which can lead to calving (Massom

et al., 2018) and contribute to sea level rise. In addition, increased wave energy paired with sea ice and permafrost loss in the Arctic could also be responsible for increased coastal erosion rates in the region (Overeem et al., 2011).

Addressing science questions related to these risks will require a variable resolution wave modeling approach. To date, global wave modeling has primarily been performed using uniform structured meshes or nested structured meshes. Unstructured triangular meshes have traditionally been limited to regional coastal applications. However, a long-term promise of unstructured

approaches is the capability to span small to large scales across the coastal to global ocean. The purpose of this paper is to report on progress toward this goal, starting with an assessment of the accuracy and performance of the WW3 model (Tolman, 1991) using global to coastal unstructured meshes. These meshes will allow wave simulations to maintain accuracy across the global and coastal oceans at reduced computational cost. This creates an opportunity to represent the effects of waves across a broad spectrum of coupled interactions within Earth system models.

The remainder of the paper is organized as follows. First, a background of variable resolution wave modeling is presented in Section 2. Second, we describe the unstructured mesh configuration we have developed for WW3 and the comparison study used to assess its accuracy and performance (Section 3). Then, the accuracy of the unstructured mesh is compared against high- and low-resolution structured meshes and measured data from buoy observations (Section 4.1 and 4.2). Next, we demonstrate the computational performance of the unstructured mesh alongside that of the structured meshes (Section 4.3). We also discuss

the results in the context of developing efficient, coastal Earth system models (Sections 5). Ultimately, we conclude that unstructured WW3 is a viable means of exploring wave interactions within coastal Earth system model applications for E3SM (Section 6).

## 2   WW3 multiresolution approaches

In this work, we analyze wave simulation accuracy and performance across the global to coastal ocean using version 6.07

of WW3 (WAVEWATCH III ® Development Group, 2019). WW3 is a third-generation spectral wave model that has been used widely for operational wave forecasting, research, and engineering applications (Chawla et al., 2013b; Alves et al., 2014; Cornett et al., 2008; Wang and Oey, 2008). Similar models include WAM (WAMDI Group, 1988) and the SWAN (Booij et al., 1999) coastal wave model. These phase-averaged wave models describe the evolution of the wave action density spectrum, $N(\lambda, \phi, k, \theta, t)$, which is a function of both longitude/latitude $(\lambda, \phi)$ and wavenumber/direction $(k, \theta)$ space. The action density

spectrum is related to the energy spectrum, $F$, by the intrinsic frequency, $\sigma$, that is observed moving with the current,

$$N(k,\theta) = \frac{F(k,\theta)}{\sigma}. \tag{1}$$





As opposed to wave energy, action density is conserved generally in the presence of ocean currents (Whitham, 1965; Bretherton and Garrett, 1968). The intrinsic frequency is given by the dispersion relationship from linear wave theory:

$$\sigma = \sqrt{gk \tanh(kd)}, \tag{2}$$

where $d$ is the depth, $k$ is the wavenumber, and $g$ is the acceleration due to gravity. The evolution of the wave action density is described by the equation:

$$\frac{\partial N}{\partial t} + \frac{1}{\cos\phi}\frac{\partial}{\partial\phi}(\dot{\phi}N\cos\phi) + \frac{\partial}{\partial\lambda}(\dot{\lambda}N) + \frac{\partial}{\partial k}(\dot{k}N) + \frac{\partial}{\partial\theta}(\dot{\theta}N) = \frac{S}{\sigma}, \tag{3}$$

where $\dot{\phi}$, $\dot{\lambda}$, $\dot{k}$, and $\dot{\theta}$ are the propagation velocities in geographic and spectral space. These propagation velocities are functions of the group velocity, ocean currents and derivatives of $\sigma$ with respect to direction. The $S$ term on the right hand side of Equation 3 is comprised of parameterized source/sink terms that represent several wave processes, i.e.,

$$S = S_{in} + S_{ds} + S_{nl} + S_{bot} + S_{db} + \dots. \tag{4}$$

These source/sink terms describe: generation due to wind ($S_{in}$), dissipation ($S_{ds}$), non-linear quadruplet interactions ($S_{nl}$), bottom friction ($S_{bot}$), and depth-limited breaking ($S_{db}$). There are several other parameterizations that can be included, (e.g., sea-ice damping, triad interactions, coastline reflection, etc.) but these are the primary terms relevant to this work.

Equation 3 requires discretization of the left hand side transport terms for numerical simulations. Traditionally, structured meshes have been employed due to the straightforward application of numerical methods and for computational simplicity (WAMDI Group, 1988). However, coastal simulations require advanced approaches. Three primary options have been developed in WW3 to address the need for variable resolution: two-way nested "mosaic" grids (Tolman, 2008), spherical multi-cell (SMC) grids (Li, 2012), and regional triangular unstructured meshes (Roland, 2008).

## 2.1 Nested and multi-cell meshes

The two-way nested mosaic approach has been extensively validated against historical wave observations and is used for NOAA forecasting operations (Chawla et al., 2013a). However, two-way nesting of structured meshes has several disadvantages (Zijlema, 2010). In these types of meshes, transitions in resolution are typically abrupt and must therefore be placed well outside regions where high resolution is needed in order to be accurate. This means high resolution regions must be larger than necessary to avoid degrading accuracy, which incurs additional computational cost. In addition, a sufficient overlap region is required to accomplish the two-way nesting, which means duplicate calculations are performed in these regions on both the coarse and fine meshes. Abrupt transitions in resolution can be mitigated by increasing the number of nested grids, at the expense of complexity. SMC grids provide an alternative multi-resolution capability. However, similar to nested meshes, SMC meshes also lack the ability to smoothly vary resolution in a flexible manner. Another option is to nest an unstructured coastal wave model, such as SWAN (Zijlema, 2010), inside a global WW3 domain (Amrutha et al., 2016). However, this approach uses the WW3 wave spectrum solution to force the boundary of the nested SWAN model, which only provides a one-way coupling between the models.





## 2.2 Triangular meshes

The previously mentioned approaches are disadvantageous for Earth system modeling because field remapping approaches
are needed to facilitate coupling between Earth system model components (Jones, 1999; Ullrich and Taylor, 2015). Field
remapping for these types of meshes is complicated, more computationally expensive, and historically has not been employed in
production Earth system simulations. Mosaic grids and WW3/SWAN nesting may be appropriate for specific regional domains
and wave modeling efforts. However, they are less applicable to global Earth system modeling due to added complexity and
heterogeneity.

In contrast, the primary advantages of triangular grids are their flexibility and ability to transition resolution smoothly
between different regions of the mesh. They also offer a more straightforward integration into coupled Earth system model
applications. To date, unstructured meshes have been primarily used in regional studies to assess accuracy in coastal settings
(Roland and Ardhuin, 2014; Abdolali et al., 2020). But, a detailed global assessment of deep to shallow water accuracy with
mesh refinement in coastal areas has not yet been performed.

The goal of this paper is to evaluate the potential for wave simulation approaches in Earth system models by demonstrating
that coastal unstructured meshes can be extended to global domains in order to efficiently simulate the wave climate across
the global and coastal ocean, from deep to shallow water. The importance of this global unstructured capability is growing
as Earth system models, e.g., E3SM (Golaz et al., 2019), are beginning to use a multi-resolution approach to understand
regional climate change impacts. Efforts to include waves into the Community Earth System Model (CESM) using structured
meshes (Li et al., 2016) have used very coarse (3x4 degree) resolution in order to keep the computational cost reasonable.
This resolution simulated the wave climate well enough to improve mixed layer depth biases in the global ocean. However,
higher mesh resolution is necessary to accurately describe coastal or high-latitude wave dynamics. This is something that is
not currently possible in structured Earth system models like CESM.

## 3  Methods

This section outlines the unstructured mesh, model configuration, and treatment of unresolved islands used in the validation
study. It also gives a description of the simulation performed and the metrics used to assess the accuracy of the unstructured
mesh results.

### 3.1  Global unstructured mesh

The unstructured mesh developed for this comparison study is designed to maintain the global and coastal accuracy of high-
resolution structured WW3 simulations at reduced computational expense. The coastal resolution is specified based on a simple
depth criteria. In this mesh, the refined resolution has been limited to U.S. coastlines in accordance with current E3SM simu-
lation campaign goals, e.g., as partially outlined in Hoch et al. (2020).





The mesh used in this study was generated using the `OceanMesh2D` software package (Roberts et al., 2019), which has shoreline resolving capabilities. Future applications will use the the `JIGSAW` mesh generator (Engwirda, 2017) once shoreline-resolving capabilities are developed for global meshes. This will allow for the same mesh generation tools to be used across the ocean, sea-ice, and wave components of E3SM. Globally, the mesh has 2 degree resolution and transitions to 1/2 degree resolution in regions where the depth is less than 4000m, as illustrated in Figure 2. Since changes in depth play an important role in the evolution of the wave field, this 4000m depth criteria has been chosen to ensure that the steep transitions from deep ocean to coastal regions and shallow inner seas are resolved (Cavaleri et al., 2018). A 10% element size grade is enforced in the transition region between the 2 degree and 1/2 degree resolutions.

The WW3 source code required minor modifications in order to enforce the periodicity of the mesh across the dateline. This minimal change corrected the edge length and area calculations for elements that straddle the -180/180 degree boundary.

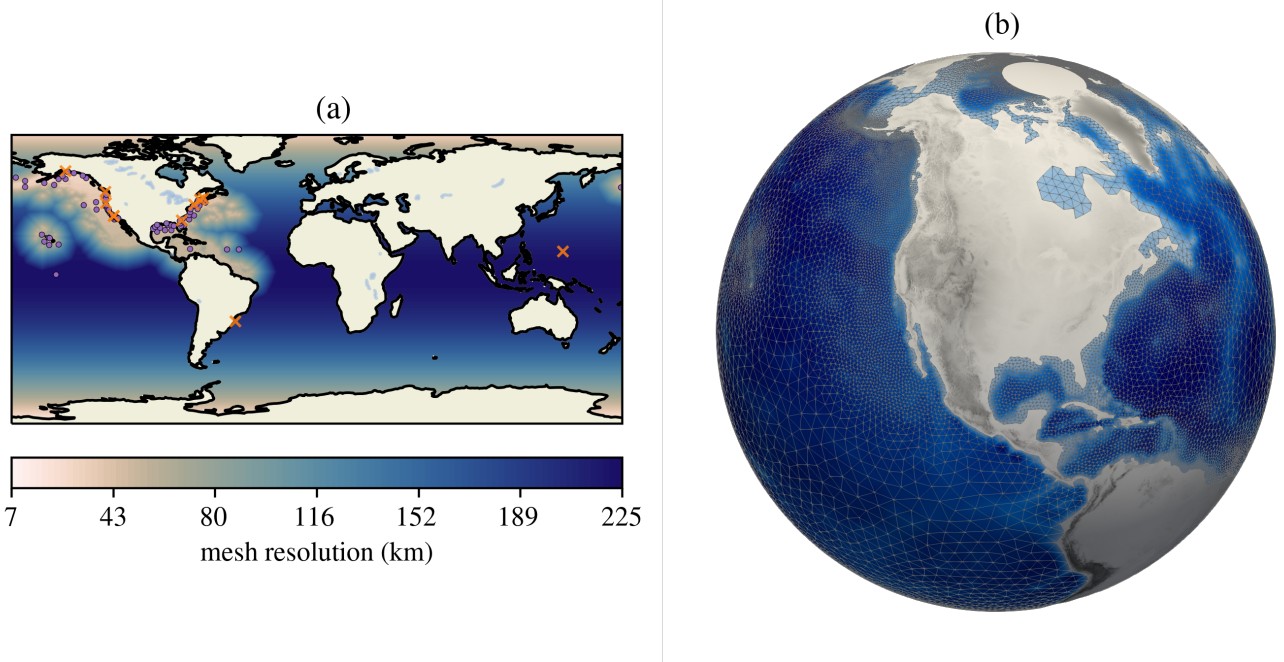

**Figure 2.** Unstructured mesh with global 2 degree resolution and half degree resolution in regions with depths of less than 4km within the coastal U.S. (a) quantitative mesh resolution, (b) qualitative mesh resolution and bathymetry. In (a), dots indicate stations used in the validation, while X's indicate those excluded as explained in Section 3.6.

## 3.2  Mesh comparisons

To validate and assess the performance of the global unstructured mesh in WW3, we have performed a study that demonstrates the accuracy and efficiency of using unstructured meshes for global applications. This was done by comparing the unstructured mesh described in Section 3.1 with 2 degree and 1/2 degree structured meshes. Both of the structured meshes were generated





**Table 1.** Mesh sizes considered in this study.

| Mesh | Size |
| --- | --- |
| 2 degree structured | 9,841 cells |
| 1/2 degree structured | 160,808 cells |
| unstructured | 16,160 nodes |

using the software developed by the NOAA WW3 development team (Chawla and Tolman, 2007). The structured meshes have been cropped at 82 degrees north due to Courant-Fredrichs-Levey (CFL) condition constraints caused by converging lines of latitude near the pole, similar to those used for NOAA operational modeling (Chawla et al., 2013a). For consistency of
comparisons, this was also done in the unstructured mesh in Figure 2b.

As we will show in the next section, the 2 degree and 1/2 structured meshes provide roughly equivalent levels of accuracy in the deep ocean. However, as expected, the 1/2 structured mesh far out-performs the 2 degree mesh in shallow coastal regions. Simulating nearshore wave dynamics with a high level of accuracy would require finer than 1/2 degree mesh resolution (Dietrich et al., 2011; Perez et al., 2017; Abdolali et al., 2020; Chawla et al., 2013a). However, we have chosen 1/2 degree as our
highest resolution since it performs reasonably well at our validation stations, and we are targeting resolutions that would be economically viable for global climate modeling applications. Note that for explicit time integration, time step restrictions due to the CFL condition for the highest resolution elements impact the overall computational time required for climate evaluations.

The goal of this comparison is to demonstrate that the unstructured mesh is able to match the deep ocean accuracy of the structured meshes, while providing equivalent accuracy to the structured 1/2 degree mesh in the refined coastal regions. Due to
having coarse resolution over most of the globe, the unstructured mesh is expected to be more computationally efficient than the global 1/2 degree structured mesh.

### 3.3 Simulation forcing

Each of the three meshes is used to perform a wave hindcast from the beginning of June 2005 to the end of October 2005. This time period represents the 2005 Atlantic hurricane season, which is the most intense hurricane season on record (Beven et al.,
2008). For the unstructured mesh this is also a challenging time period because strong seasonal swell from the Southern Ocean during these months (Young, 1999) is generated in coarse regions of the mesh and propagates into the refined region. Since wind-wave dynamics represent a damped system forced by atmospheric winds, i.e., they have a short memory response, long hindcast periods are of lesser importance for validation purposes. The model is forced using winds from the Climate Forecast System Reanalysis (CFSR) (Saha et al., 2010) at 1/2 degree spatial resolution and hourly time intervals. In this study, no sea ice
concentration or ocean reanalysis data is used in the simulations. The bathymetry dataset for all three meshes is the ETOPO1 bathymetry product (Amante and Eakins, 2009), with global coverage and sufficient resolution to represent islands.





**Table 2.** WW3 physics switches used.

| Switch | Description | Reference |
|--------|-------------|-----------|
| ST4 | Generation and dissipation (e.g., whitecapping) | Ardhuin et al. (2010) |
| DB1 | Depth-limited breaking | Battjes and Janssen (1978) |
| BT1 | JONSWAP bottom friction | Hasselmann et al. (1973) |
| NL1 | Discrete interaction approximation (DIA) for quadruplets | Hasselmann et al. (1985) |
| UOST | Unresolved islands | Mentaschi et al. (2015) |

## 3.4 Model configuration

The source/sink term combinations from Equation 4 can be selected in WW3 using different model "switches". The options used for both the structured and unstructured mesh simulations are shown in Table 2. We use the ST4 source terms by Ardhuin et al. (2010) since they have been shown to have lower significant wave height biases as compared to other source term packages (Stopa et al., 2016). Standard approaches are used for bottom friction, non-linear quadruplet interactions, and depth limited breaking (Chawla et al., 2013a). As discussed in the next section, we also include an unresolved islands source term approach instead of the discretization-level correction commonly used in structured mesh WW3 studies (Tolman, 2003). For each of the switches mentioned, we have used their default parameter settings.

The spectral grid is the same as used in Chawla et al. (2013a). It consists of 36 directions and 50 frequencies with a frequency range of 0.036-0.963Hz (corresponding to a frequency interval of 1.07). The same spectral mesh is used across all geographic meshes considered.

In order to be consistent across the structured and unstructured meshes, the standard explicit fractional timestepping scheme is used. The maximum timesteps are as follows: 900s global, 300s geographic, and 450s spectral. The minimum source term timestep is 30s. This set of timesteps is used for all geographic meshes and is consistent with the finest resolutions used. An implicit method is available for unstructured meshes, but it is not considered here.

Typically, the third-order ULTIMATE QUICKEST scheme (PR3 and UQ switches) is used for the structured mesh discretization. However, the first order PR1 propagation scheme is used for the structured meshes in this study in order to make a fair comparison against the first order CRD-N scheme used for the unstructured mesh (Roland, 2008).

## 3.5 Unresolved islands

One of the key sources of error in global wave models is the missing dissipation due to unresolved islands. Two different approaches have been developed to include these sub-grid scale effects. The first is implemented as a correction factor to the numerical flux between cells in the propagation scheme (Tolman, 2003). The correction factors are calculated based on the fraction of a cell that is obstructed by an island. This approach is specific to the structured grid propagation schemes in WW3 and has not been generalized to unstructured grids.



The second approach is based on a source term that accounts for the effects of unresolved islands (Mentaschi et al., 2015). This source term considers both the local dissipation and the shadow effect of upstream cells. One advantage of this approach is that it can be used for both structured and unstructured grids. The coefficients required for this source term can be calculated using the open-source python package `alphaBetaLab` (Mentaschi et al., 2019).

In our validation and analysis, we have used the source term approach for both the structured and unstructured model configurations and we focus on accuracy in shallow coastal regions. Not only has the source term parameterization been shown to be slightly more accurate for the structured meshes, but it also allows for a consistent approach across the unstructured and structured simulations (Mentaschi et al., 2018). Thus, differences between results are due to the use of a structured or unstructured mesh and grid resolution. All other factors have been kept constant to make direct comparisons between results,

to within minor interpolation errors.

### 3.6  Buoy validation dataset

Validation is performed using best available data sets and community-accepted approaches. Altimeter-based validation has already been presented for global unstructured meshes in WW3 (Mentaschi et al., 2020) and is therefore not repeated here. Instead, we make direct comparisons against available buoy data that spans the global and coastal ocean. Buoy data is commonly

used to directly quantify the accuracy of spectral wave models, especially for coastal areas. Large-scale systematic biases, which may not necessarily be quantified via sparse and regional buoy data, are assessed against the 1/2 degree structured mesh that has already been validated for the global ocean (Chawla et al., 2013a). Validation protocols are detailed below.

In Section 4.1, data from each of the three meshes is compared with significant wave height data measured by buoys from the National Buoy Data Center (NCBC) (Meindl and Hamilton, 1992), between June-October 2005 and shown at the locations

in Figure 2a. These buoys give an hourly time history of significant wave height, dominant wave period, peak frequency, wind speed, and wind direction. All active buoys from this time period have been used in this analysis, except for 11 out of 115 buoys that either experienced instrument failures or were outside the primary study region. The stations that have been used (○) and those that were excluded (×) are shown in Figure 2a. We have focused on significant wave height because it provides the largest dataset across the buoy measurements, compared to frequency and direction observations that are more difficult to

observe accurately (Steele et al., 1998). In our analysis, we exclude the first week of the simulation to account for the spin-up period.

### 3.7  Validation metrics

We use several different metrics to assess the errors between our modeled results and the buoy observations. The first is the commonly used root mean square error (RMSE) defined as:

$$RMSE = \sqrt{\frac{1}{N}\sum_{i=1}^{N}(M_i - O_i)^2},$$    (5)





where $M_i$ is the modeled result, $O_i$ is the observed value and $N$ is the number of observations. We also show distributions of the bias error

$$e_i = M_i - O_i, \quad i = 1, \ldots, N \tag{6}$$

and relative (normalized bias) error

$$\delta_i = \frac{M_i - O_i}{O_i}, \quad i = 1, \ldots, N. \tag{7}$$

These error distributions are plotted by binning each $e_i$ and $\delta_i$ value throughout the simulation. The counts are then normalized to account for differences in temporal observational coverage for each station. Presenting the errors in this way gives a richer description of the accuracy of a given model than averaged error quantities that can misrepresent skewed biases.

In Section 4.2, we also compute average solution differences against the structured 1/2 degree mesh over the entire global
domain to access spatial solution variability due to resolution. These differences are computed by interpolating the unstructured and 2 degree structured solutions onto the 1/2 degree mesh. The average and maximum of the $e_i$ and $\delta_i$ values are computed from global 3-hourly significant wave height model output.

## 4 Results

In this section we describe the results of our validation with buoy data and show global differences between the 1/2 degree
structured mesh and the 2 degree structured and unstructured meshes. We also give an assessment of computational performance.

### 4.1 Buoy data validation

Significant wave height error comparisons are shown in Figures 3 - 10. Each of these figures represents a different set of stations based on geographical location. For each region, subplot (a) shows the RMSE for the station locations in subplot (b).
The normalized distribution of the error and relative error for each station can be found in subplots (c) and (d), respectively.

Comparing the error and relative error distributions can reveal whether the largest errors occur at the largest or smallest observed wave height conditions. If the error distribution becomes more peaked, i.e., the tails are eliminated in the relative error distribution, the largest errors occur at large observed wave heights. In the opposite case, i.e., the error distribution is more diffused compared to the relative error distribution, the larger errors occur during smaller observed wave heights.
In each figure, the station depths are sorted from deep to shallow to depict accuracy differences at varying ocean depths and mesh resolutions. Generally, in the deep ocean, all mesh resolutions provide the same solution accuracy as implied by Li et al. (2016). However, in shallow coastal areas, the high resolution provided by the 1/2 degree structured and unstructured meshes is necessary to maintain the same accuracy level obtained for the deep ocean.

Quantitative evaluations of significant wave heights are presented corresponding to the wave buoys in 8 different regions.
These include: the Gulf of Maine (Section 4.1.1), South- to Mid-Atlantic East Coast (Section 4.1.2), Gulf of Mexico (Section



4.1.3), Caribbean Region (Section 4.1.4), Southern California Coast (Section 4.1.5), Northern California and Pacific Northwest Coast (4.1.6), Alaskan Coast (Section 4.1.7), and Hawaiian Coast (Section 4.1.8). A summary of the relative errors for the three meshes across the deep and shallow buoys in these regions can be found in Table 3, which broadly summarizes the success of the unstructured mesh for both deep and shallow ocean wave simulation. A more detailed analysis for each region is detailed 260 in the following subsections.

### 4.1.1 Gulf of Maine

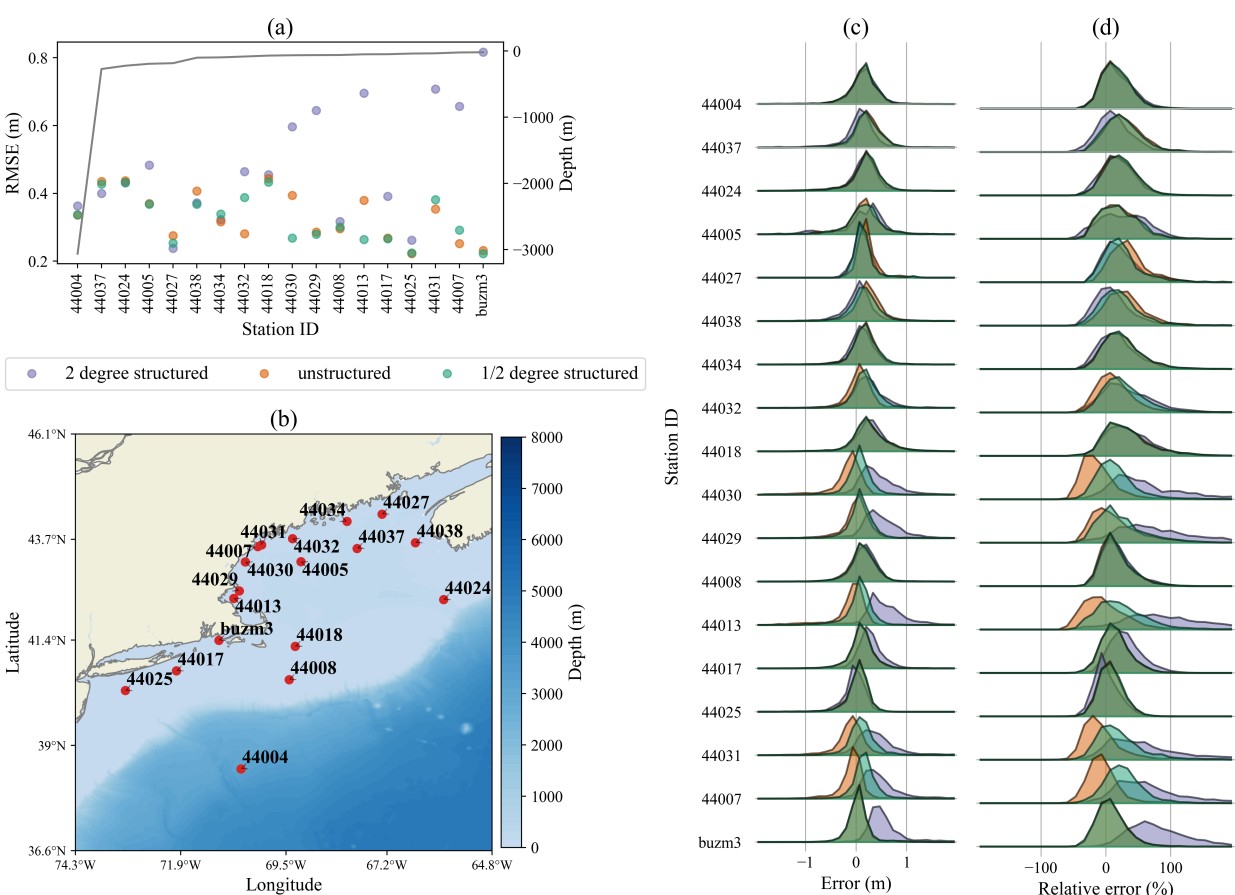

**Figure 3.** Comparison between modeled results and NDBC buoy data for the 2 degree structured, unstructured and 1/2 degree structured meshes in the Gulf of Maine region. (a) Root mean squared errors (dots) for each mesh resolution along with station depth (grey line). (b) Geographic location of each station. (c) Normalized distribution of bias errors between the model and observations over the simulated time period. (d) Normalized distribution of relative errors.

In Figure 3, the unstructured mesh is of comparable accuracy to the 2 degree and 1/2 degree grids for stations 44004-44034 on the $x$ axis of (a). The unstructured mesh is more accurate than both the structured meshes at station 44032. For stations





44030-buzm3, the unstructured mesh provides substantial accuracy improvements over the 2 degree mesh and its quality
approaches 1/2 degree results, albeit with smaller amplitudes for some buoy locations. The exceptions in this range are stations
44008 and 44025, which are shallow stations further from the coast where all models provide similar accuracy.

### 4.1.2   South to Mid-Atlantic East Coast

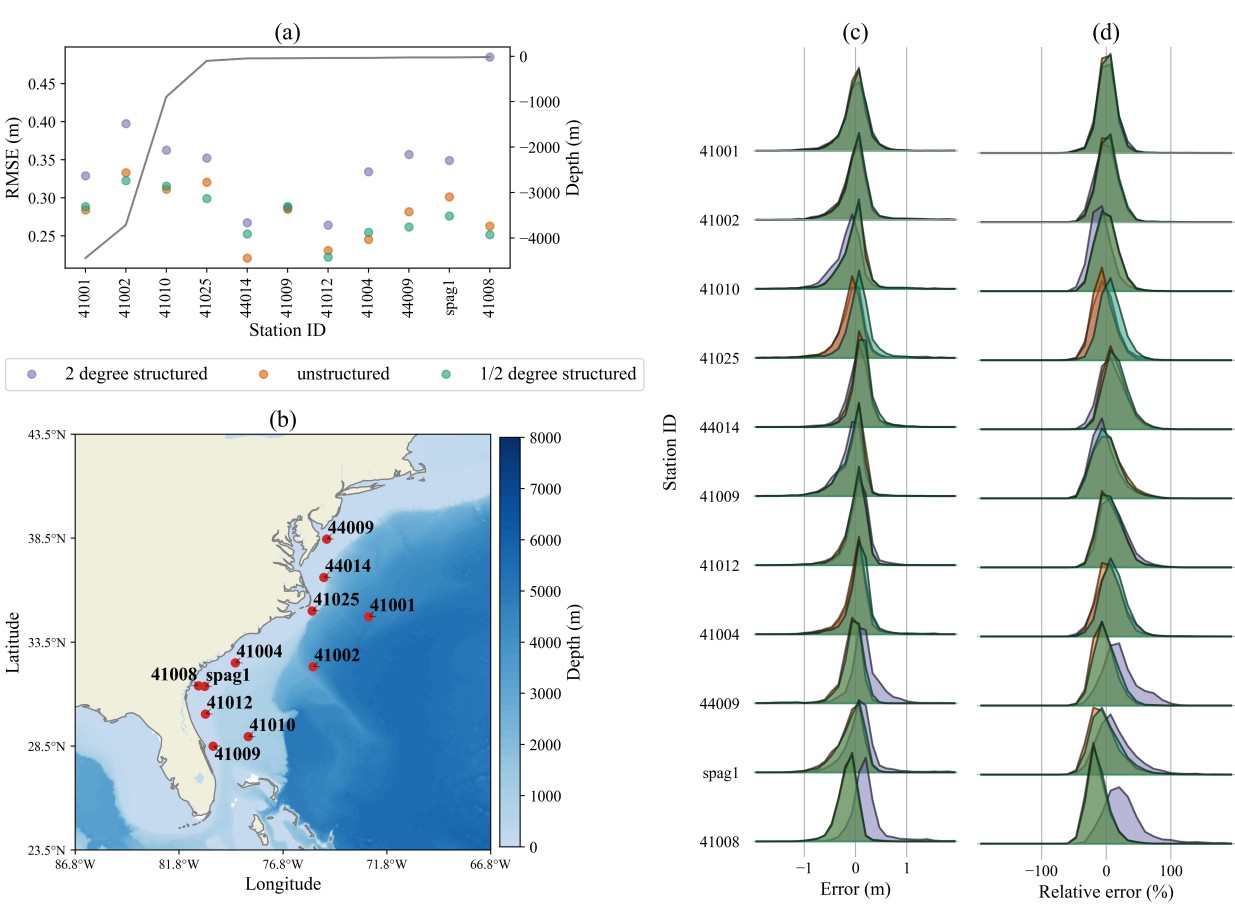

**Figure 4.** Same as Figure 3 for stations in the south to mid-Atlantic East Coast.

Further south of this region, Figure 4 shows that the unstructured mesh solution generally achieves similar accuracy to the
1/2 degree structured grid and improves upon the 2 degree structured grid. The representation of the shelf break likely plays a
role in the accuracy of the unstructured mesh, as exemplified by stations 44014 and 41025. The unstructured mesh provides the
most accurate result at 44014, but does not improve as drastically over the 2 degree solution at a similarly placed shelf-break
station to the south, at station 41025.





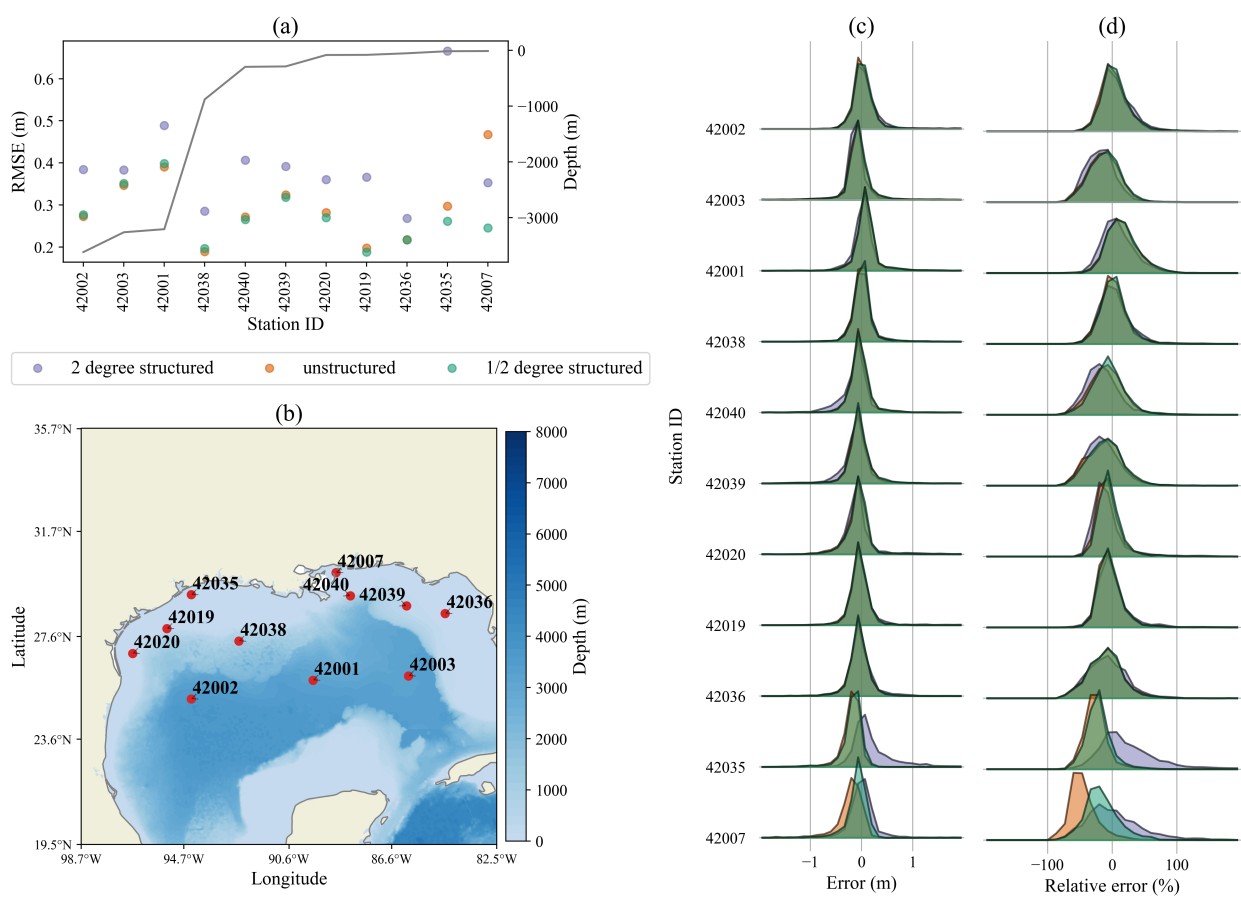

**Figure 5.** Same as Figure 3 for stations in the Gulf of Mexico.

### 4.1.3 Gulf of Mexico

In the Gulf of Mexico, shown in Figure 5, the unstructured and 1/2 degree results agree well and improve upon the 2 degree
solution for all but one station. The biggest improvement for the unstructured mesh over the 2 degree mesh is at station 42035,
where the 2 degree mesh is the most inaccurate. The worst station for the unstructured mesh is at 42007. This occurs because
the unstructured mesh begins to resolve the "bird's foot" portion of the Mississippi River delta, shielding the wave energy
at this station. Throughout the Gulf of Mexico, the unstructured mesh has nearly uniform 1/2 degree resolution, illustrating
parity between the 1/2 degree structured and 1/2 degree unstructured approaches. Furthermore, waves in this region are locally
generated within the Gulf. Therefore, good agreement between the unstructured and 1/2 degree solutions is expected. This
demonstrates that the unstructured mesh performs as well as the equivalent resolution structured mesh in wind-sea dominated
conditions.




### 4.1.4 Caribbean Region

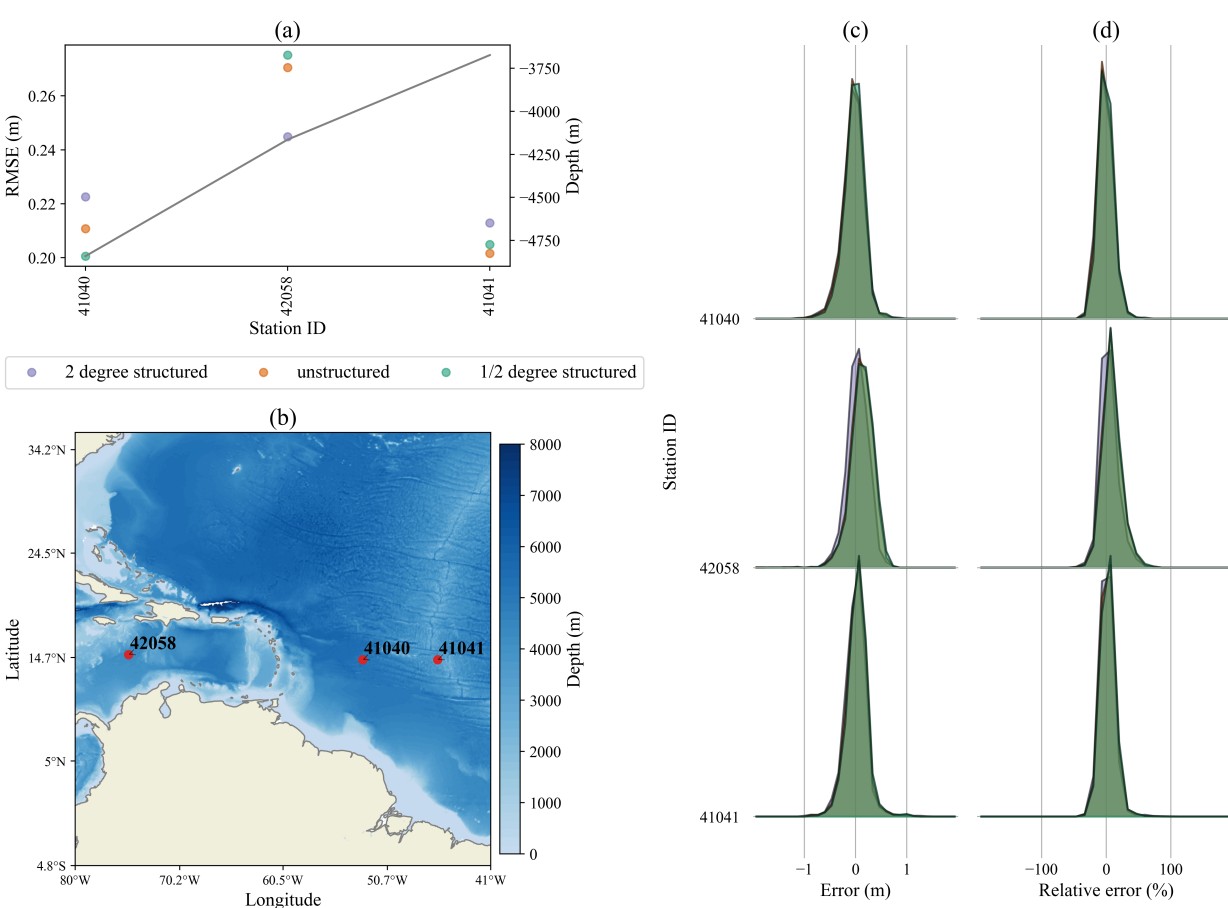

**Figure 6.** Same as Figure 3 for stations in the Caribbean Sea and Atlantic Ocean.

In the Figure 6 stations, the overall RMSE is the lowest of all the regions considered. As a result, the differences between
the meshes are exaggerated when compared to the other (a) figures. All three meshes perform very similarly at these stations
for both the Atlantic and Caribbean Sea locations, which have different levels of sheltering via land.

### 4.1.5 Southern California Coast

As shown in Figure 7, the unstructured mesh outperforms the 2 degree mesh at most of the stations in the southern U.S.
west coast. This is a challenging region to model at the resolutions considered here, due to the strong swell environment and
presence of several small islands near the coast. In terms of RMSE, the unstructured grid generally performs as well as the 1/2
degree mesh. However, overall, the unstructured grid solution tends to under-predict the 1/2 degree and 2 degree structured
grids, which both have consistent positive biases. The more negative bias of the unstructured mesh in this region leads to



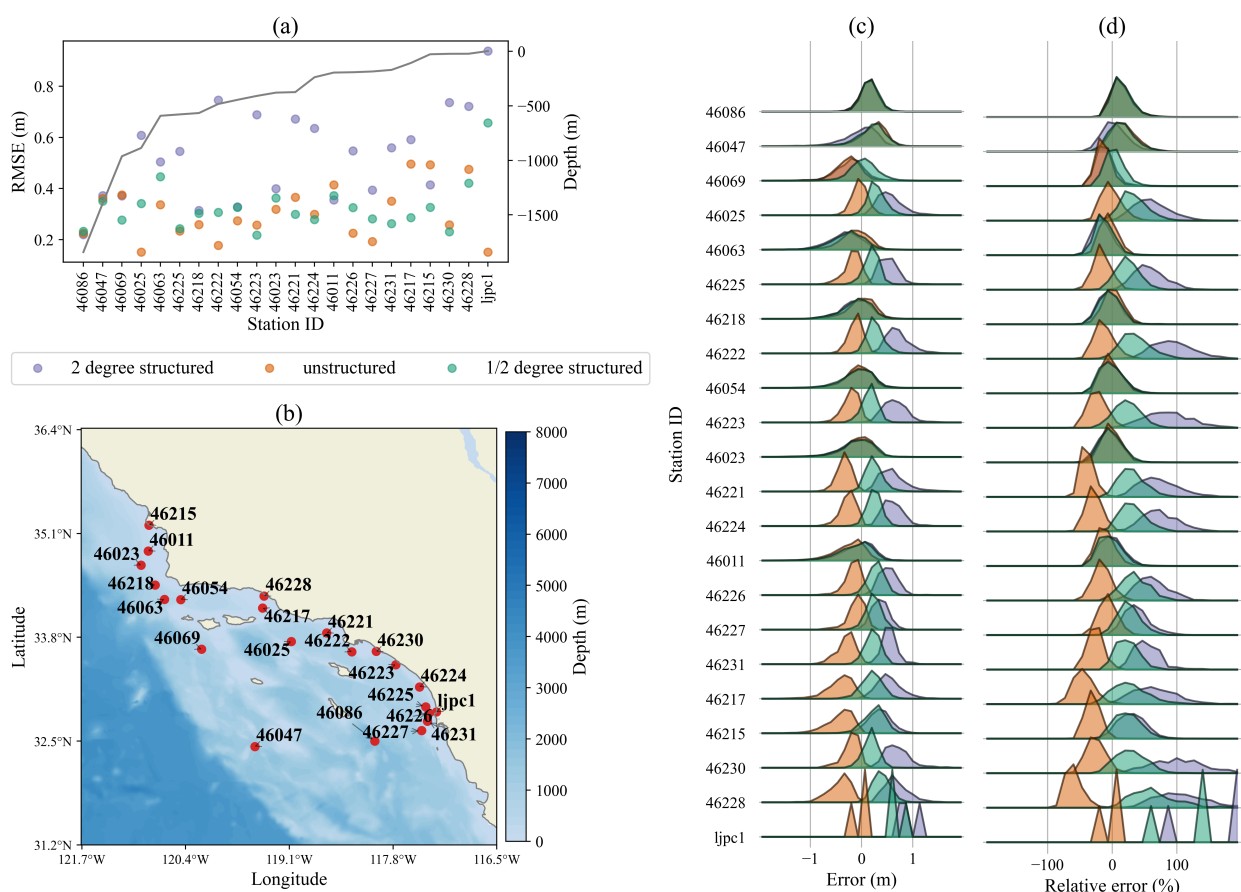

**Figure 7.** Same as Figure 3 for stations along the southern California coast.

more accurate results at some stations. In panel (c), stations 46025, 46222, 46226, and 46227 for the unstructured mesh all under-predict the 1/2 degree grid, but have error distributions that are centered more toward zero. There are, however, several

stations where the unstructured grid under-performs. At station 36215, for example, the unstructured mesh is the least accurate solution. Since all of the offshore islands in this region are represented by the unresolved obstacle source term, the solution in this region is highly sensitive to the accuracy of the sub-grid island parameterization. In nearly all of the unstructured stations, the relative error distributions are more concentrated about their mean than the bias error distribution. This indicates that higher errors are occurring during large wave height conditions, leading to lower relative errors. The opposite tends to be true for the

2 degree resolution, as the relative error distribution generally has a greater spread than the bias error distributions.

### 4.1.6  Northern California and Pacific Northwest Coast

Figure 8 shows the buoy validation results for the northern west coast of the U.S. As with the other west coast stations in Figure 7, this coast experiences a swell-dominated wave climate. This region has a very narrow shelf, which is difficult to adequately



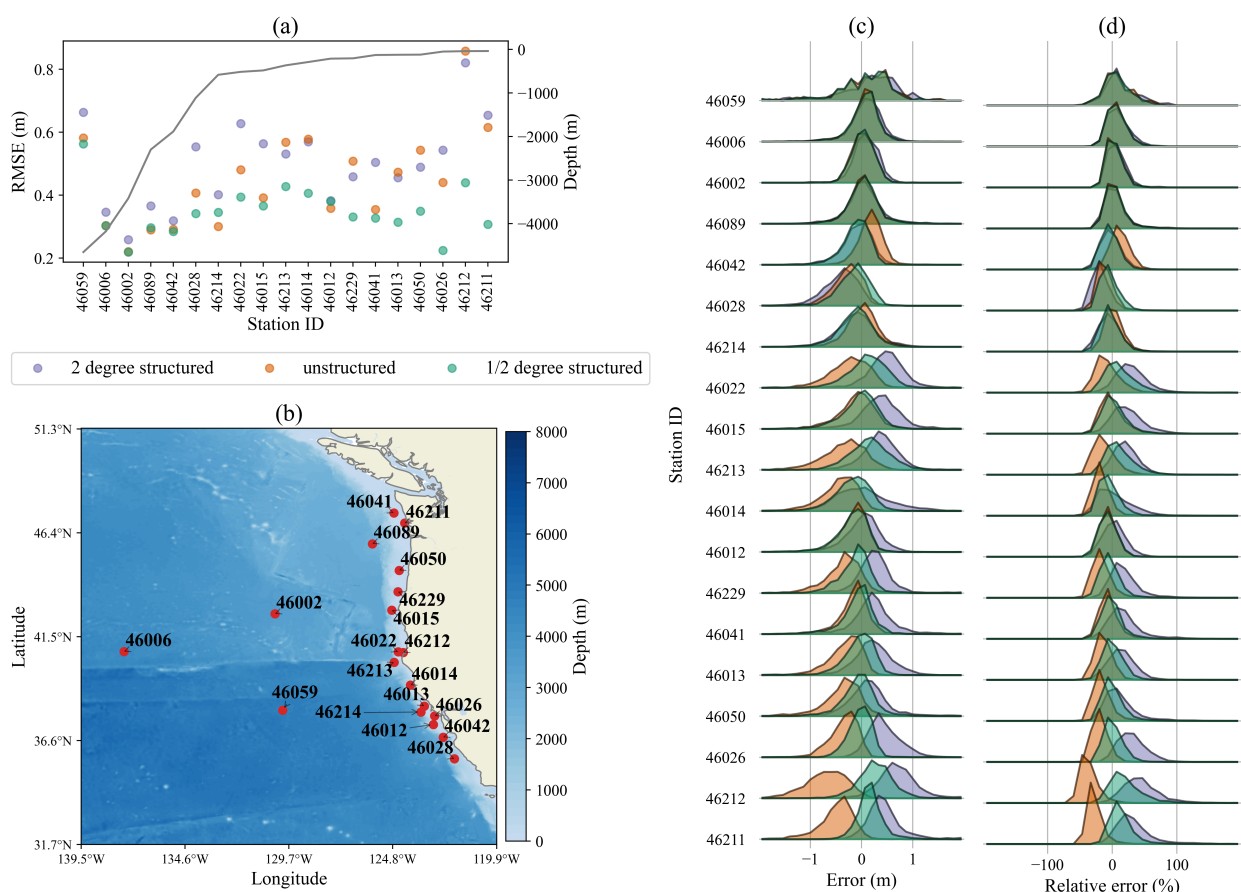

**Figure 8.** Same as Figure 3 for stations along the northern California and Pacific Northwest coast.

resolve at 1/2 degree resolution. Again, the unstructured mesh results typically under-estimate both of the structured mesh solutions. All models perform similarly at the deep water stations. The unstructured mesh has some difficulties in this region; it has similar RMSE values as the 2 degree mesh solution at stations 46213, 46014, 46229 46013, 46050, 46212, and 46211. This could possibly be mitigated by providing additional resolution along the coastline to better resolve the narrow shelf in this region. The tails of the error distributions for the unstructured mesh tend to be eliminated in the relative error distributions. Again, this indicates that the largest errors occur during the largest observed wave heights.

### 4.1.7 Alaskan Coast

For the Alaska region shown in Figure 9, the models again perform similarly at the deep stations with the unstructured and 1/2 degree meshes having a slight accuracy advantage. The unstructured mesh shows a large improvement over the 2 degree mesh at station 46060, but it is the least accurate solution at the nearby station 46061. Other than these two stations, the unstructured mesh agrees well with the 1/2 degree structured grid.





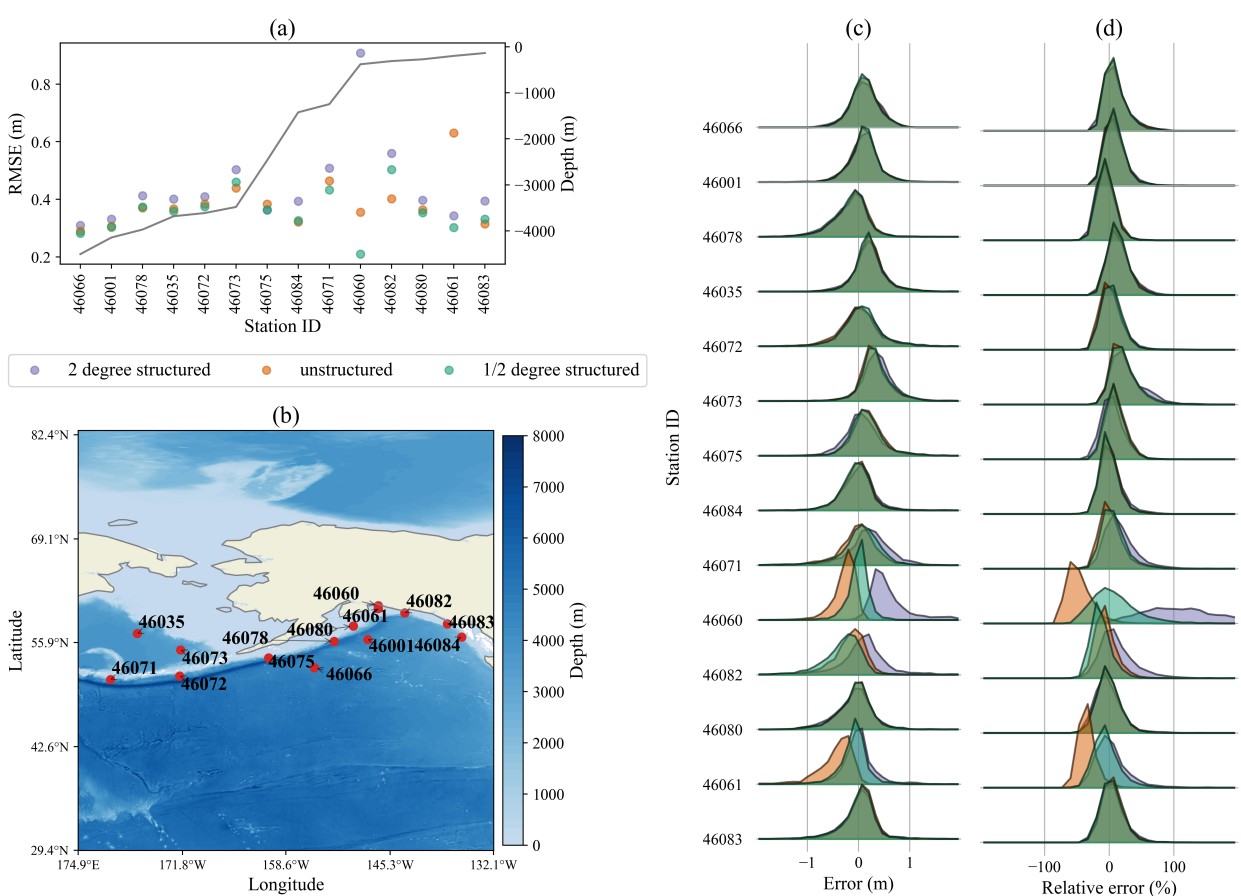

**Figure 9.** Same as Figure 3 for stations in the Alaskan coastal region.

### 4.1.8 Hawaiian Coast

The final station region considered is around the Hawaiian Islands as shown in Figure 10. The deep water stations again show equivalent accuracy for all three mesh resolutions. Near the island of O'ahu, the unstructured mesh is the most accurate solution at station 51201 and the least accurate at 51202. The Island of Hawaii is the only Hawaiian Island that is explicitly resolved in the mesh, so the results at these two stations are heavily dependent on the unresolved source term parameterization.

### 4.2 Global solution differences

The results from the previous section show that, overall, the 1/2 degree structured mesh provides the most accurate significant wave height results – although, the unstructured results are often comparable, as illustrated at many stations. Taking the 1/2 degree structured results as the "reference" solution, we now show global averaged and maximum solution differences against the 1/2 degree structured mesh plotted for the 2 degree structured and unstructured solutions. Of course, this comparison

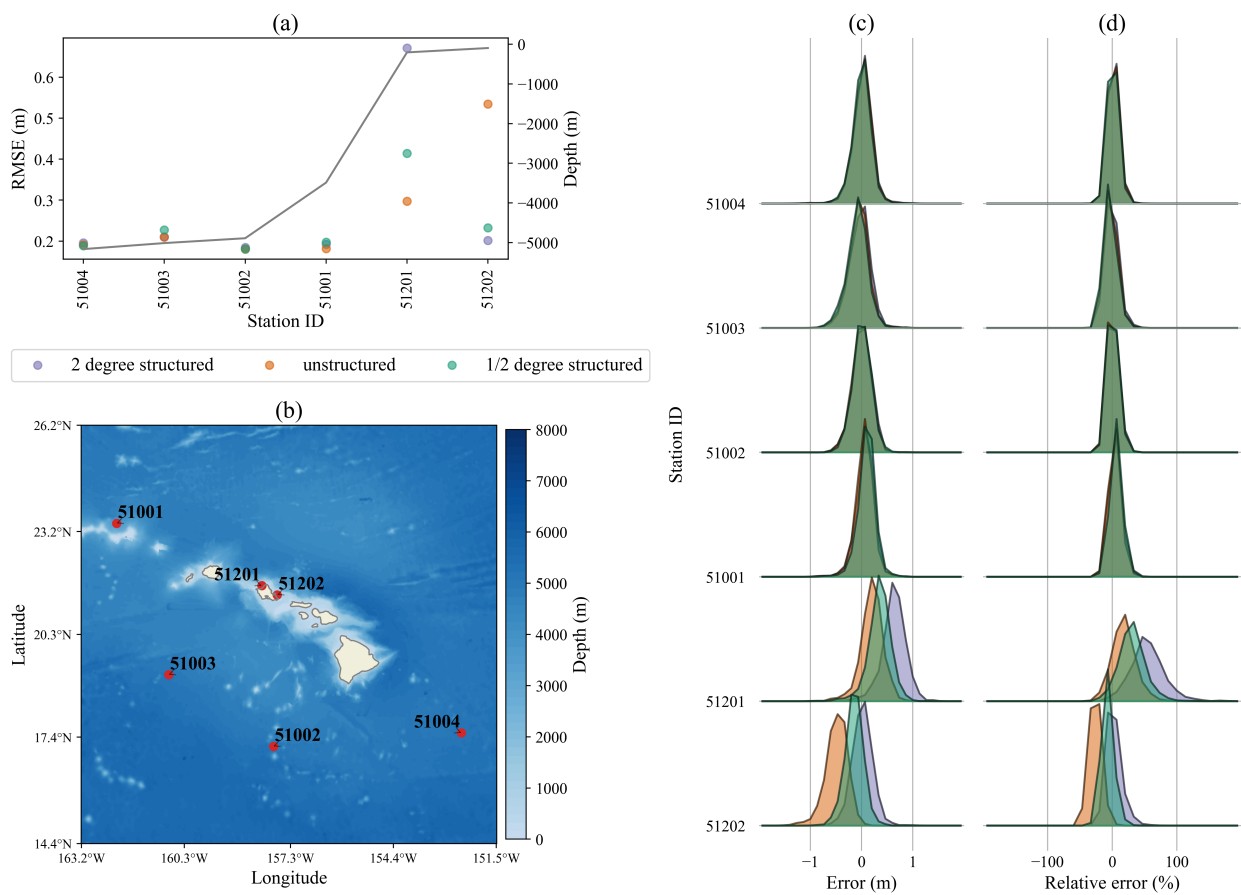

**Figure 10.** Same as Figure 3 for stations near the Hawaiian islands.

neglects inherent biases of the modeled results with respect to observations. However, it demonstrates spatially that the deep
ocean accuracy is largely unaffected by increased resolution, while in the refined coastal regions, the unstructured mesh is in
better agreement with the 1/2 degree structured mesh.

Figure 11 shows that in the deep ocean, the unstructured and 2 degree meshes have similar differences compared to the 1/2
degree mesh. Especially in the equatorial regions, the solutions for these grids compare well to the 1/2 degree solution in terms
of both mean and maximum differences. The unstructured mesh has larger mean and maximum differences in the Southern
Ocean, especially south of Africa. Figure 12 shows the average and maximum relative differences for the 2 degree structured
and unstructured meshes compared to the 1/2 degree structured mesh. The large discrepancies in the Southern Ocean found in
Figure 11, are not as apparent on a relative basis because large waves are present in that area year-round. The mean absolute
relative difference between the 1/2 structured mesh for both the 2 degree structured and unstructured meshes is around 5% over
most of the deep ocean basins.





In the refined coastal areas of the unstructured mesh, the solution provides a much better agreement to the 1/2 degree structured mesh in terms of both mean and maximum differences. Generally, improvements for the mean are better expressed by the absolute differences in Figure 11, while the improvement in the maximum differences can be seen more clearly on a relative basis as shown in Figure 12.

In particular, the Gulf of Mexico benefits the most from the regional refinement of the unstructured mesh. In this area, average relative differences of around 10% are reduced to around 3% in the unstructured mesh. Again, the 1/2 structured mesh and unstructured mesh have the same resolution throughout this area, so good agreement is expected for the wind sea dominated conditions. The U.S. East Coast also benefits from the unstructured refinement, especially in terms of absolute maximum differences. The same is true for the maximum absolute differences in the Gulf of Alaska.

Note, differences from individual high wind events appear to be "imprinted" in the difference plots. This is due to the relatively short June-October time period used in this study. A longer simulation period would produce a more even distribution of maximum differences. Another factor that contributes to these large maximum differences is the low resolution of the wind forcing for the 2 degree structured mesh and in the coarse regions of the unstructured mesh. The 2 degree resolution cannot accurately capture storm systems, leading to a more diffuse wind field. Therefore, larger differences in maximum significant

wave height are associated with the lower-fidelity representation of intense wind events.

There are also greater coastal differences in the 2 degree portions of the unstructured mesh as compared to the 2 degree structured mesh. This is because the unstructured mesh better conforms to the coastlines at this resolution, which gives it more overlap with the 1/2 degree structured mesh in these regions. However, due to the coarse 2 degree resolution, the accuracy in these shallow coastal areas suffers in the unstructured mesh. The combination of these factors leads to bands of high error

that are not present in the 2 degree structured mesh because of reduced overlap with the 1/2 degree structured mesh in coastal regions.

### 4.3   Performance and scalability

A strong scaling study was performed to assess the efficiency gains of the unstructured mesh. For each of the three meshes, a series of one month runs were performed using between 36 and 1800 processors. All point, field, and restart file output

was turned off to directly compare the time required to compute the solution. Each timing is based on an average of three runs. The timing simulations were performed on the Grizzly cluster, which is a part of Los Alamos National Laboratory's Instititional Computing Program. Each Grizzly node has 128GB of memory and is composed of two 18 core E5 2695v4 Broadwell processors, each with a clock speed of 2.1GHz. The network is built on the Intel Omni-Path Architecture with 100Gbps bandwidth.

The parallelization strategy for structured WW3 meshes is based on a "card-deck" approach (Tolman, 2002). In this approach, the cells are distributed to processors in a round-robin fashion. The source term calculation and intra-spectral propagation are computed in parallel and the data is gathered onto a single core to perform the propagation in geographic space. This method incurs more data communication than a traditional domain decomposition approach. However, it has been shown to scale and has the advantage that the same parallel scheme can be used regardless of the spatial propagation method.

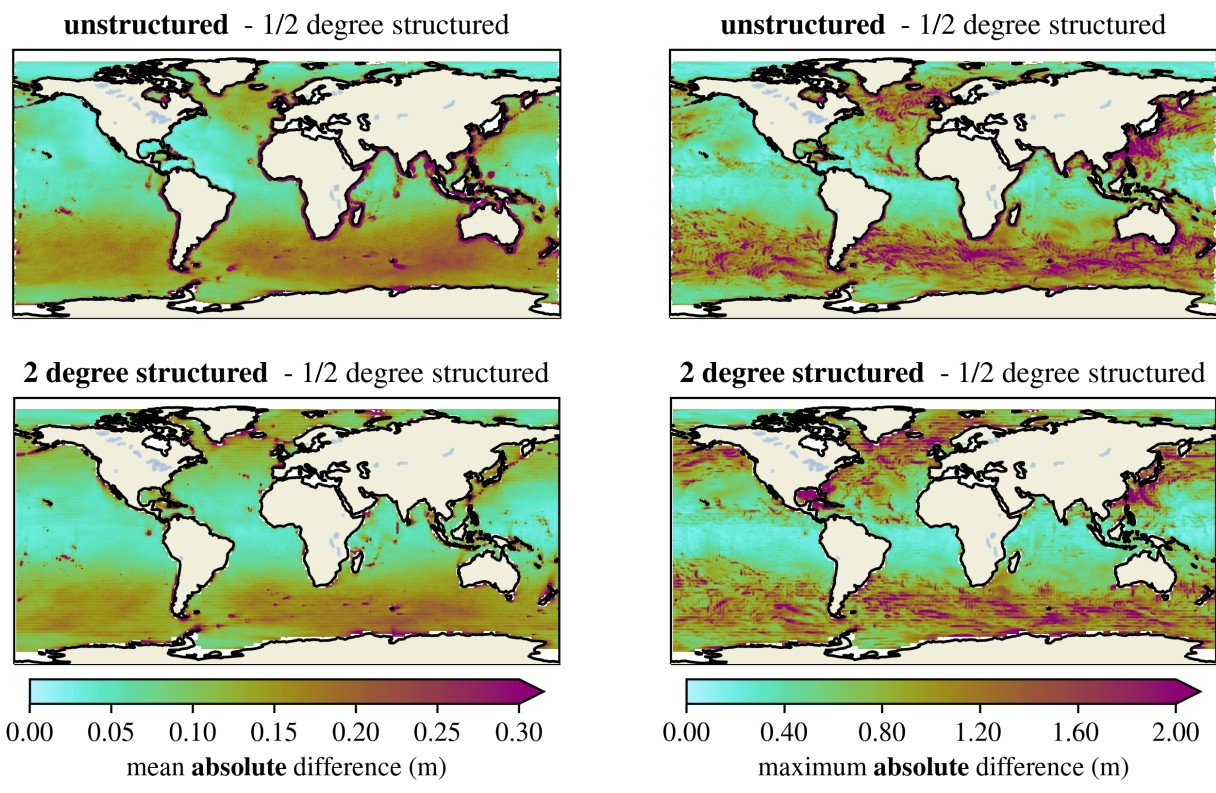

**Figure 11.** Mean (left panel) and maximum (right panel) absolute differences in significant wave height between the unstructured and 1/2 degree structured meshes (top row) and the 2 degree and 1/2 degree structured meshes (bottom row).

A traditional domain decomposition method has also been developed for the unstructured grids. In our comparison, we have chosen to use the card-deck approach to fairly compare across the structured and unstructured mesh configurations because domain decomposition is not available for the structured meshes in the existing WW3 source code. A comparison of the card-deck and domain decomposition approaches can be found in Abdolali et al. (2020).

     Figure 13 shows the scaling results for each of the three mesh resolutions. The 1/2 degree structured mesh scales nearly
ideally until 576 cores, and begins to experience a slowdown after 1152 cores. The unstructured mesh begins to scale less than ideally after 72 cores because of the reduced size of the mesh, but continues to experience a speedup until 576 cores. Because it has the smallest number of cells, the 2 degree mesh does not scale after 72 cores. Even with its improved scaling, the 1/2 degree mesh does not achieve faster run times than the unstructured mesh. It is 2.2 times slower than the unstructured mesh at 1152 cores, which is the point at which both meshes stop experiencing a speedup with increasing core count. Due to the
scaling advantage of the unstructured mesh over the 2 degree mesh, the runtimes between the two meshes become nearly equal at 1152 cores. The throughput in terms of simulated years per day peaks at 12.1, 10.9, and 4.9 for the 2 degree, unstructured, and 1/2 degree meshes, respectively.



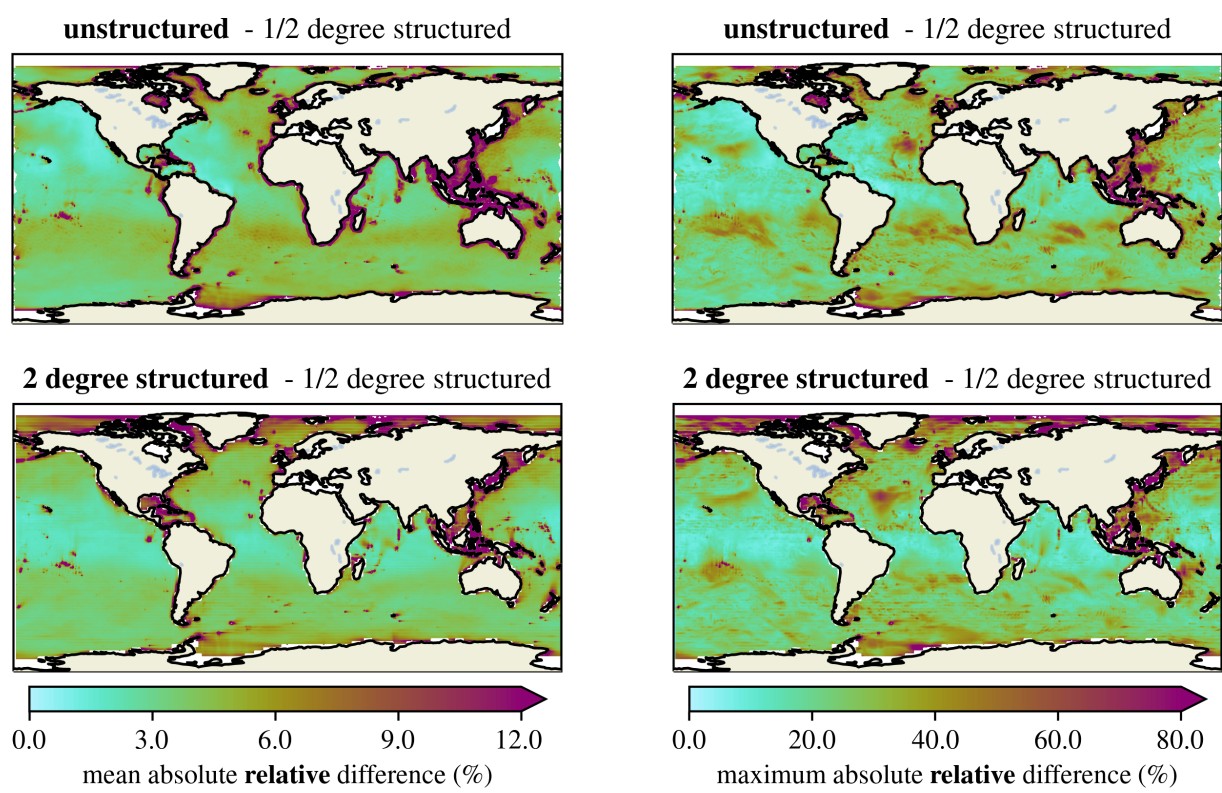

**Figure 12.** Mean (left panel) and maximum (right panel) absolute *relative* differences in significant wave height between the unstructured and 1/2 degree structured meshes (top row) and the 2 degree and 1/2 degree structured meshes (bottom row). Note this is different from Figure 11, which shows absolute differences.

## 5 Discussion

In general, our results show that unstructured meshes can be used globally to match the accuracy of high-resolution uniform

structured meshes in regions of interest at reduced cost. Figure 12 demonstrates that mean relative differences between 2 degree and 1/2 degree resolution in the deep ocean are generally less than 5%. This means the benefit to having increased resolution in these regions is minimal. However, Table 3 shows that the 1/2 degree coastal resolution of the unstructured mesh provides substantial accuracy improvements over the 2 degree structured mesh in shallow regions. In some cases, the unstructured mesh reduces the average relative errors in coastal regions by half, as compared to the 2 degree mesh. This puts the coastal accuracy of

the unstructured mesh near that of the 1/2 degree mesh, but for lower computational cost. In terms of computational efficiency, the unstructured mesh is between 2-10 times faster than the 1/2 structured mesh, depending on the number of cores used.

Another advantage of this approach is that global unstructured meshes are more flexible and are simpler to configure than two-way nested structured grids. The emergence of flexible and robust open-source mesh generation tools such as





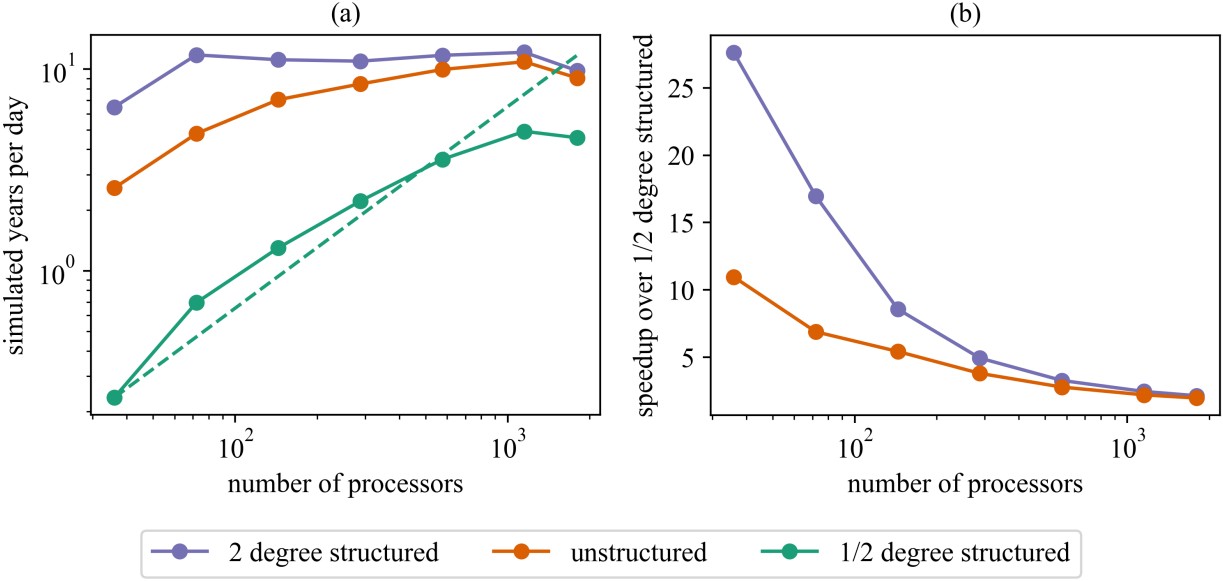

**Figure 13.** (a) Throughput in terms of simulated years per day based off a 1 month of simulation with each mesh. The dashed line represents ideal scaling for the 1/2 degree structured grid. (b) speedup for the unstructured and 2 degree meshes over the 1/2 degree structured grid for each core count.

**Table 3.** Comparison of average absolute relative errors between deep (depth > than 1000m) and coastal (depth ≤ 1000m) stations for each mesh. The regions shown correspond to those found in Figures 3-10

| | Deep | | | Coastal | | |
|---|---|---|---|---|---|---|
| Region (section) | 2 degree | unstructured | 1/2 degree | 2 degree | unstructured | 1/2 degree |
| Gulf of Maine (4.1.1) | 22.5 | 20.3 | 20.9 | 52.2 | 25.8 | 28.0 |
| S. to Mid-Atlantic E. Coast (4.1.2) | 13.2 | 11.7 | 11.9 | 19.8 | 16.0 | 16.3 |
| Gulf of Mexico (4.1.3) | 20.7 | 20.2 | 20.0 | 26.2 | 22.3 | 19.8 |
| Caribbean Region (4.1.4) | 9.3 | 9.5 | 9.5 | - | - | - |
| S. CA Coast (4.1.5) | 15.7 | 18.5 | 17.9 | 51.5 | 22.1 | 25.9 |
| N. CA and Pacific N.W. Coast (4.1.5) | 14.8 | 12.7 | 11.6 | 25.2 | 18.5 | 14.6 |
| Alaskan Coast (4.1.7) | 16.5 | 14.3 | 14.3 | 41.7 | 24.8 | 17.5 |
| Hawaiian Coast (4.1.8) | 7.4 | 7.0 | 7.4 | 31.9 | 23.6 | 20.8 |

`OceanMesh2D` (Roberts et al., 2019) and `JIGSAW(GEO)` (Engwirda, 2017) have reduced the difficulty that was previ-
ously associated with generating unstructured meshes. These tools allow for arbitrary resolution specification and produce
high-quality meshes that can be used directly in simulation.





There are, however, some challenges associated with unstructured meshes in practice. One of these issues is the representation of the coastlines with increasing resolution in coastal regions. Figure 14 shows the coastline differences between the unstructured and 1/2 degree meshes. As the coastline resolution in the unstructured mesh begins to resolve details of the coast, the resulting coastline geometry can lead to local inaccuracies. For example, single-element islands can be more accurately described using the unresolved obstacle terms than by being explicitly represented in the mesh. Another example is near the Mississippi River Delta in the Gulf of Mexico. Here, the delta is just beginning to be resolved, which leads to an exaggerated sheltering effect that is not present in the structured mesh. The mesh could be altered to mitigate these issues but this was not explored here for consistency and to avoid accidental "mesh tuning" of the results.

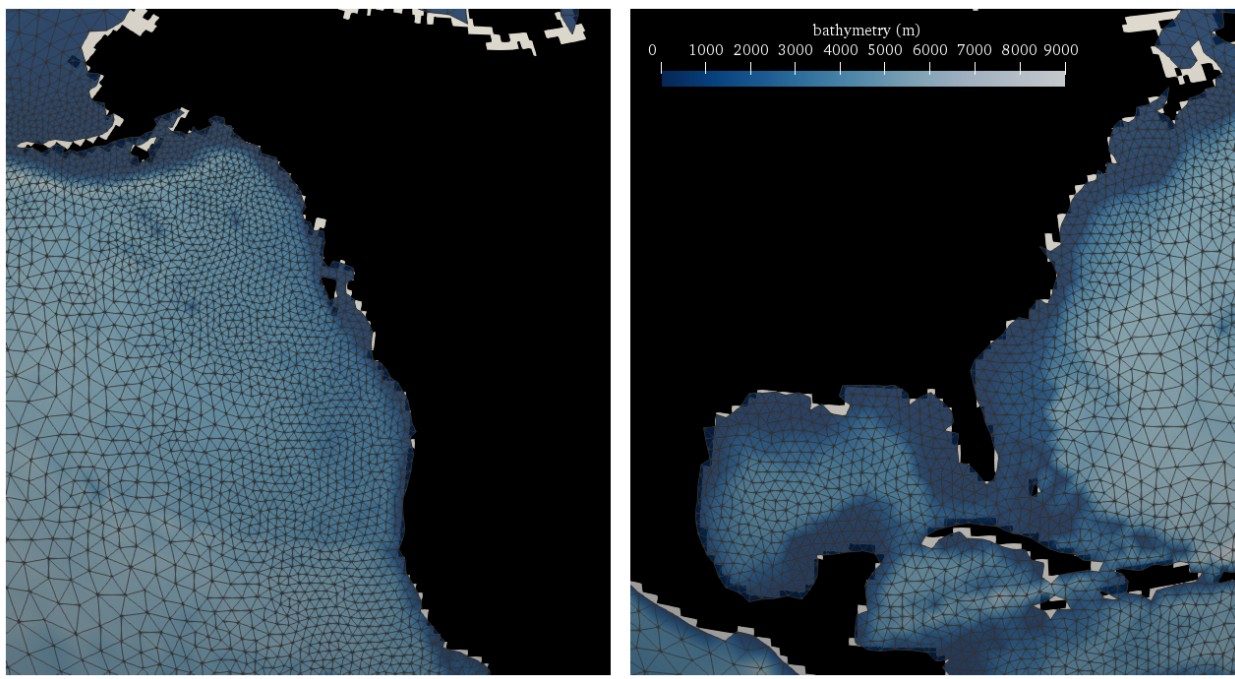

**Figure 14.** Differences in coastline representation between the 1/2 degree structured and unstructured meshes for the U.S. West Coast (left) and U.S. East Coast/Gulf of Mexico (right). The structured mesh coastline is represented in black. The white regions and overlaps between the black coastline and the triangular elements indicate mismatches in the mesh boundaries.

Sub-grid modeling of islands and other unresolved obstacles is a critical source term needed to achieve good results with the unstructured mesh. Figure 15 shows the large errors that are obtained by excluding this source term for both the open ocean and coastal regions. The major advantage of the source term approach is that it can be applied to the unstructured mesh, unlike the discretization-level approach that is implemented for structured grids in WW3.

Additional performance gains are expected to be possible using the unstructured mesh via implicit timestepping methods, which are not as useful for structured meshes. The implicit method is advantageous for meshes that vary drastically in resolution



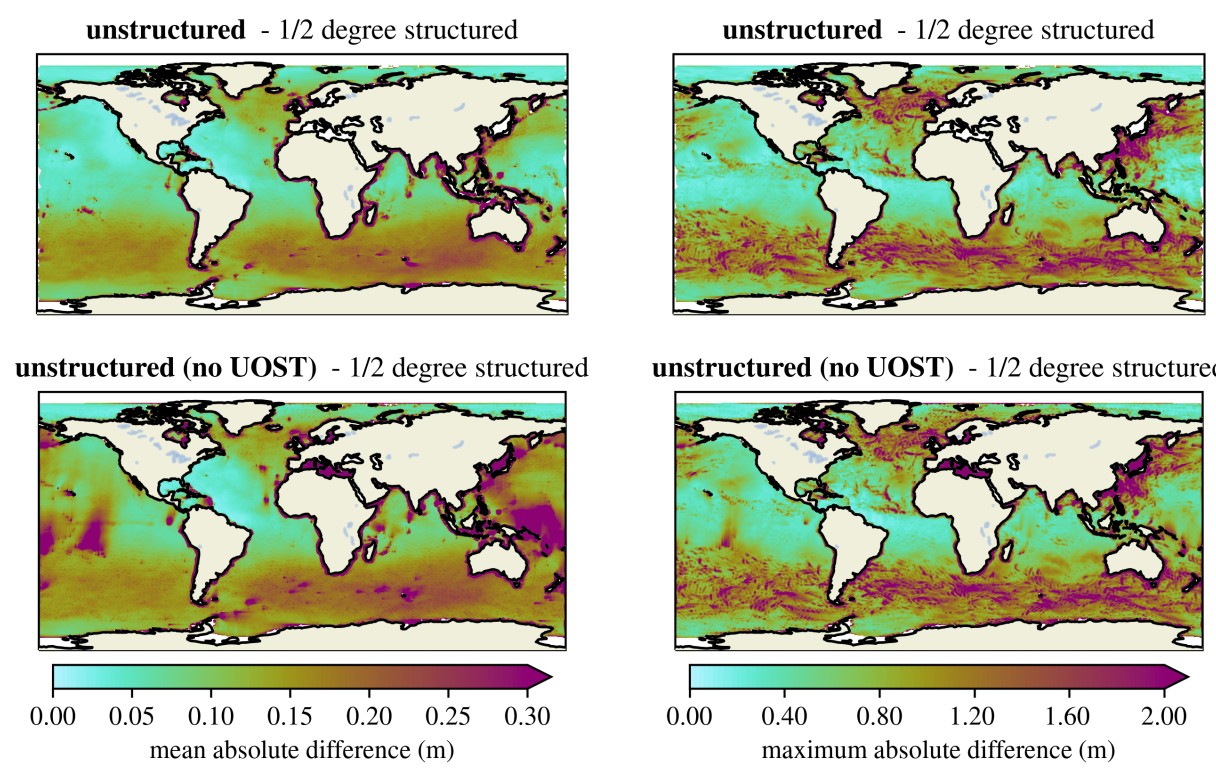

**Figure 15.** Mean (left panel) and maximum (right panel) absolute differences in significant wave height between the unstructured mesh with the UOST and 1/2 degree structured grid (top row) and between the unstructured mesh without the UOST and 1/2 degree structured mesh (bottom row).

since the smallest element size no longer dictates the maximum stable timestep. A comparison between the explicit and implicit approaches can be found in (Abdolali et al., 2020).

The unstructured mesh presented here is based on a simple, depth-based refinement criteria that transitions between 2 degree and 1/2 degree resolutions. A generic mesh was used to make a clean comparison between structured meshes with those two grid spacings, not to optimize for accuracy. This approach details the potential of unstructured meshes to provide accuracy across the global and coastal scales, even for general meshing approaches. However, unstructured meshes also provide flexibility to consider other mesh design criteria or provide more focused resolution in specific regions. The results presented here could be improved in particular regions via targeted mesh design criteria. Overall, the unstructured mesh seems to perform very well in coastal wind-sea dominated environments, such as the Gulf of Mexico and U.S. East Coast. Swell-dominated environments, such as the U.S. West Coast and Alaska tend to have larger errors for the unstructured case. Ultimately, unstructured meshes can accommodate global to coastal resolution across various coastal environments without sacrificing accuracy or efficiency.



# 6 Conclusions

Our results demonstrate the accuracy and efficiency advantages of global unstructured meshes. By varying resolution across
deep and shallow regions, unstructured meshes are able to match the efficiency of coarser resolution structured meshes in the
global ocean without sacrificing accuracy in high resolution coastal areas. This capability has the potential to greatly reduce
the computational cost of including waves into Earth system models.

The validations presented show that in coastal regions, the unstructured mesh considered here was as accurate as the 1/2
degree structured mesh. The unstructured mesh provided a 2-10 times speedup over the global 1/2 degree structured mesh
depending on the number of cores used. This increase in performance comes with substantial coastal accuracy gains over the
global structured 2 degree results.

The unstructured WW3 capability combined with the variable resolution philosophy of E3SM is a viable approach to in-
cluding waves in coupled Earth system simulations. This allows interfacial interactions between the atmosphere and ocean to
be directly simulated via inclusion of wave physics in WW3. The performance of this approach is expected to allow for consid-
eration of an expanded range of science applications including coastal flooding, coastal biogeochemistry, and wave-sea ice in-
teractions at high latitudes. Inclusion of wind-wave processes, especially for coastal simulations, is needed for next-generation
Earth system models as resolution and overall complexity increase. These interactions between coupled components will begin
to require resolution of sub-grid processes dependent upon wave conditions as directly simulated by WW3. The advantages of
unstructured WW3 make it ideal for use in next-generation, variable-resolution Earth system models like E3SM.

*Code and data availability.* Model setup files for the 2 degree structured, unstructured, and 1/2 degree structured grids can be accessed at
Brus et al. (2020a, e, c), respectively. The simulation results and observed data are achieved at Brus et al. (2020d). The code used in this
work can be found at Brus et al. (2020b).

*Author contributions.* Steven Brus: Conceptualization, Methodology, Software, Analysis, Visualization, Writing–original draft. Phillip Wol-
fram: Conceptualization, Methodology, Analysis, Writing–review and editing. Luke Van Roekel: Conceptualization, Methodology, Writing–
review and editing. Jessica Meixner: Conceptualization, Writing–review and editing.

*Competing interests.* The authors declare no competing interests

*Acknowledgements.* Initial development of the unstructured mesh framework for coastal flooding was funded under the Los Alamos Na-
tional Laboratory Research and Development Directed Research project "Adaption Science for Complex Natural-Engineered Systems"
(20180033DR). The present work was funded under the Energy Exascale Earth System Model (E3SM) project, funded by the U.S. Depart-





450  ment of Energy Office of Science, Office of Biological and Environmental Research. Computing resources were provided by Los Alamos
National Laboratory Institutional Computing , US DOE NNSA (DE-AC52-06NA25396).





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
