# Peer review of "Unstructured global to coastal wave modeling for the Energy Exascale Earth System Model using WAVEWATCH III version 6.07"

_Geoscientific Model Development, 2020_

## Referee Comment (RC1) · Anonymous Referee #1 · 4 Jan 2021

General comments

The manuscript is well-written and the authors set out their objectives and results very clearly. It is suitable for publication in GMD if the following comments are addressed.

Specific comments

Line 25: why are phase-averaged wave models quite expensive compared to the atmospheric and oceanic dynamic models? Do you have evidences for this? By the way, are you talking about "spectral wave models" or "phase-averaged" (which is more general, e.g. XBeach).

Line 113: please add reference Dietrich et al (2011) for completeness.

Line 188: the first order PR1/CRD-N schemes have been employed. To what extent do they influence the outcomes (i.e. less accurate) in particular the 2-degree mesh case?

Technical corrections, etc.

Line 295: 36215 -> 46215

---

## Referee Comment (RC2) · Anonymous Referee #2 · 24 Jan 2021

Review:

This is a groundbreaking and well-written study. The authors demonstrate how an unstructured WAVEWATCH III (WW3) spectral wave model setup can be used to conduct variable-resolution global simulations, with purposes of coupling with an earth system model. Like the two-way nested structured mesh approach, the unstructured model resolves coastal areas with similar accuracy of a higher resolution uniform-structured-mesh model but with lower computational costs. The study uses a consistent method to determine global mesh refinement. The unstructured approach is straight-forward to integrate with earth system models, and is an ideal choice for coupling with variableresolution earth system models such as the E3SM.

General comments:

A1. The first half of Section 2.1 (L91-98) seems to imply that using a two-way nested structured mesh setup in WW3 is more computationally expensive than a variable resolution unstructured mesh (in terms of achieving similar accuracy). Is this a correct interpretation of the text? And if so, is it more of an opinion or is it based on previous studies or work by the authors?

I ask because regional wave climate studies often use WW3 with a nested structured mesh setup to resolve coastal regions and readers might be curious how this approach compares with the unstructured mesh setup.

My understanding is that there is a higher computational overhead (or at least was initially) when using an unstructured mesh (as well as differences in accuracy). Depending on the setup, it seems highly plausible that a two-way nested model could be more accurate or faster than using the unstructured approach (despite calculating potentially overlapping cells).

While a comparison of these two approaches is likely outside the scope of this study, it seems like a potentially missing component to make any definitive statements on accuracy or cost.

A2. As a follow up, I am wondering if this manuscript has overlooked situations where a nested structured mesh setup with WW3 might be preferable to an unstructured mesh for global simulations. Of course, it seems quite natural and desirable to use WW3 with an unstructured mesh with variable resolution earth system models such as the E3SM. However, there are some earth system models that do not use variable resolutions (e.g., ocean circulation model with a fixed horizontal spatial resolution) but do employ high-resolution atmospheric models (e.g., 0.2 degree spectral grid). In such situations, it seems like the nested structured mesh approach might be better since

communication with the atmospheric component would require limited field remapping (assuming near-coincident mesh placement).

In light of this, it might be worthwhile to review a few statements, such as those found in L47, L109, and L424-426, to see if any minor clarifications are necessary.

A3. One of the strengths of using the unstructured approach in global simulations is highlighted nicely by the sentence on L413-414.

Could the authors elaborate on why 4000 m was chosen for the refinement? In this wave model setup, the largest wave length resolved is around 1200 m, and depth effects are not likely to become important until around 600 m or shallower. I imagine the choice has to do with ensuring smooth transitions but it would be nice to know more details.

A4. In the comparisons with buoy data, there seems to be a negative bias at many of the stations when using the unstructured mesh model. I can see why the 2 degree structured mesh model might have a positive bias but the former case is not so clear to me. This may be more of an open question but do the authors have an idea of why this might be occurring (e.g., differences in numerical scheme)? Is this just a feature for this particular setup or does the negative bias persist for different resolution refinements (e.g., unstructured mesh with a 2 degree to a 1 degree transition)?

As a potential follow up, what is the beta parameter value used for the ST4 physics? In Rascle and Ardhuin (Ocean Modelling, 2013), the beta value is tuned for use with NCEP CFSR winds using a structured mesh setup. It seems plausible that similar improvements could be made tuning the same parameter with an unstructured mesh setup.

A5. L418-420 & Table 3: Can you clarify the following statements in L418-420?

"Overall, the ... have larger errors for the unstructured case."

Is this referring to the buoy validation results or global comparisons with the 0.5 degree

structured mesh? In the latter case, it does appear that the biggest improvements in agreement coincide with the Gulf of Mexico and the U.S. East Coast. However, in Table 3, there are dramatic performance increases in coastal regions of the Gulf of Maine, Alaska, and S. CA Coast when using the unstructured WW3. Aside from the N. CA and Pacific N. W. Coast, this improvement tends to be in regions with significant swell.

Specific comments:

B1. L188-189: How much does the use of the PR1 propagation switch affect accuracy for the structured meshes? This seems like important information to include since most structured mesh setups would not likely use this setting.

B2. Section 4.1: What model values are being used for the buoy station comparisons? Is a nearest neighbor approach being used (i.e., using the cell average computed by WW3 that the station falls within)? And if not, how sensitive are the results to interpolation method?

B3. L265-266: Is buoy station 44025 relatively far from the coast compared with other the buoy stations?

B4. L269: I recommend changing "The representation" to "Better representation", or similar.

B5. Table 3: It's a little odd that the 2 degree structured WW3 model performs slightly better in deep water than the 0.5 degree model for the S. CA Coast. Do the authors have any ideas why this might be?

Minor comments:

C1. I believe the proper nomenclature for the model (in the title and manuscript) should be 'WAVEWATCH III'.

C2. L151-152: Please add 'degree' to both instances of "1/2 structured mesh/es".

C3. L295: Station should be '46215' and not '36215'.

C4. Table 3: Please state that table values are percentages.

References:

N. Rascle, F. Ardhuin, A global wave parameter database for geophysical applications. Part 2: model validation with improved source term parameteri- zation, Ocean Model. 70 (2013) 174-188

---

## Referee Comment (RC3) · Anonymous Referee #2 · 25 Jan 2021

The unstructured mesh presented here refines U.S. coastal regions at around 0.5 degrees using around 16,000 nodes. Does anyone know approximately how many nodes would be necessary to refine coastal regions for the entire globe?

---

## Author Comment (AC1) · 10 Mar 2021

**Response to Reviewer #1**

**The manuscript is well-written and the authors set out their objectives and results very clearly. It is suitable for publication in GMD if the following comments are addressed.**

Thank you for reading through our manuscript and for providing valuable comments. Please find our point-by-point responses to your comments below. Reviewer comments are in bold and author responses are in plain text.

**Specific comments:**

Line 25: why are phase-averaged wave models quite expensive compared to the atmospheric and oceanic dynamic models? Do you have evidences for this? By the way, are you talking about "spectral wave models" or "phase-averaged" (which is more general, e.g. XBeach).

The expense of the wave model is due to the need to evolve the frequency/direction spectrum of the wave action in longitude/latitude space. This leads to a greater number of overall degrees of freedom compared to an ocean or atmospheric models. In the case of our study, the spectral mesh resolution means that there are 1800 unknowns per grid point (50 frequencies  $\times$ 36 directions), as compared to a typical low-resolution E3SM configuration where the ocean and atmosphere models have less than 100 vertical layers at each grid point. The low-resolution E3SMv1 model ran at 10 simulated years per day (Golaz et al., 2019), which is roughly what we found the throughput of the unstructured mesh solution to be in this study. For additional evidence, we also point to the study done in Li et al. (2016) where WAVEWATCH III coupled to the CESM model. In order to reduce cost relative to the rest of the coupled model, the wave model resolution used in that paper was very coarse:  $3.2 \times 4$ degree with a 25 frequency and 24 direction spectral grid.

Also, in the submitted manuscript we did use "spectral wave model" and "phase-averaged" wave mode interchangeably. Based on your comment, we now realize that it is more accurate to refer to spectral wave models as a specific class of phase-averaged model. We have done our best to clear this up in the text by referring to WAVEWATCH III as a "*spectral, phase-averaged wave model*"

**Line 113: please add reference Dietrich et al. (2011) for completeness.**

We have added this reference, as suggested.

**Line 188: the first order PR1/CRD-N schemes have been employed. To what extent do they influence the outcomes (i.e. less accurate) in particular the 2-degree mesh case?**

Thank you for raising this point. We have done our best to address this in Figure 1. This figure shows that the standard PR3+UQ schemes for structured meshes are more accurate than the first order PR1 switch. However, the differences do not alter the conclusions of our study. We have added this figure as an Appendix in the manuscript.

**Technical corrections :**

**Line 295: 36215 -> 46215**

Thank you for carefully reading through the text to catch this mistake. We have corrected it.

**References**

- Golaz, J.-C., Caldwell, P. M., Van Roekel, L. P., Petersen, M. R., Tang, Q., Wolfe, J. D., Abeshu, G., Anantharaj, V., Asay-Davis, X. S., Bader, D. C., et al.: The DOE E3SM coupled model version 1: Overview and evaluation at standard resolution, Journal of Advances in Modeling Earth Systems, 11, 2089–2129, 2019.
- Li, Q., Webb, A., Fox-Kemper, B., Craig, A., Danabasoglu, G., Large, W. G., and Vertenstein, M.: Langmuir mixing effects on global climate: WAVEWATCH III in CESM, Ocean Modelling, 103, 145–160, 2016.

---

## Author Comment (AC2) · 10 Mar 2021

**Response to Reviewer #2**

This is a groundbreaking and well-written study. The authors demonstrate how an unstructured WAVEWATCH III (WW3) spectral wave model setup can be used to conduct variable-resolution global simulations, with purposes of coupling with an earth system model. Like the two-way nested structured mesh approach, the unstructured model resolves coastal areas with similar accuracy of a higher resolution uniform-structured-mesh model but with lower computational costs. The study uses a consistent method to determine global mesh refinement. The unstructured approach is straight-forward to integrate with earth system models, and is an ideal choice for coupling with variable-resolution earth system models such as the E3SM.

Thank you for reading through our manuscript and for providing thoughtful comments. They were very useful in helping us improve the message of this paper. Please find our point-by-point responses to your comments below. Reviewer comments are in bold, author responses are in plain text, and modifications to the manuscript are in italics.

**General comments:**

A1. The first half of Section 2.1 (L91-98) seems to imply that using a two-way nested structured mesh setup in WW3 is more computationally expensive than a variable resolution unstructured mesh (in terms of achieving similar accuracy). Is this a correct interpretation of the text? And if so, is it more of an opinion or is it based on previous studies or work by the authors? I ask because regional wave climate studies often use WW3 with a nested structured mesh setup to resolve coastal regions and readers might be curious how this approach compares with the unstructured mesh setup. My understanding is that there is a higher computational overhead (or at least was initially) when using an unstructured mesh (as well as differences in accuracy). Depending on the setup, it seems highly plausible that a two-way nested model could be more accurate or faster than using the unstructured approach (despite calculating potentially overlapping cells). While a comparison of these two approaches is likely outside the scope of this study, it seems like a potentially missing component to make any definitive statements on accuracy or cost.

That is the correct interpretation of the text, but, as of now, it is based more on educated conjectures than previous studies or recent work by the authors. We agree that more direct comparisons with nested grids and SMC are out of scope of this current study, but would be required for definitive statements on accuracy or cost. To not imply direct comparisons and better reflect our scope, Section 2.1 was rewritten as follows:

The two-way nested mosaic approach has been extensively validated against historical wave observations and is used for NOAA forecasting operations (Chawla et al., 2013a). However, two-way nesting of structured meshes has several disadvantages (Zijlema, 2010). In these types of meshes, transitions in resolution are typically abrupt. Therefore, they must either be placed well outside regions where high resolution is needed in order to be accurate or a series of nested meshes must be employed to

achieve a smooth transition. This means high resolution regions must be larger than necessary to avoid degrading accuracy or the complexity of the nested model. Two-way nesting also requires a sufficient overlap region, which means duplicate calculations are performed in these regions on both the coarse and fine meshes. SMC grids provide an alternative multiresolution capability. However, similar to nested meshes, SMC meshes also lack the ability to smoothly vary resolution in a flexible manner. Another option is to nest an unstructured coastal wave model, such as SWAN (Zijlema, 2010), inside a global WW3 domain (Amrutha et al., 2016). However, this approach uses the WW3 wave spectrum solution to force the boundary of the nested SWAN model, which only provides a one-way coupling between the models. For the E3SM coupled application, the two-way feedback from coastal to the global wave model is desired.

A2. As a follow up, I am wondering if this manuscript has overlooked situations where a nested structured mesh setup with WW3 might be preferable to an unstructured mesh for global simulations. Of course, it seems quite natural and desirable to use WW3 with an unstructured mesh with variable resolution earth system models such as the E3SM. However, there are some earth system models that do not use variable resolutions (e.g., ocean circulation model with a fixed horizontal spatial resolution) but do employ high-resolution atmospheric models (e.g., 0.2 degree spectral grid). In such situations, it seems like the nested structured mesh approach might be better since communication with the atmospheric component would require limited field remapping (assuming near-coincident mesh placement). In light of this, it might be worthwhile to review a few statements, such as those found in L47, L109, and L424-426, to see if any minor clarifications are necessary.

We agree and acknowledge that the multi-grid nested case and the SMC grid case were not considered in this study. We limited the investigation to structured meshes and unstructured triangle meshes. Minor clarifications were made to better express this:

L 53: The purpose of this paper is to report on progress toward this goal, starting with an assessment of the accuracy and performance of the WW3 model (Tolman, 1991) using global to coastal unstructured meshes, which are compared to a single structured grid.

L108-109: However, they are not considered here due to their added complexity and heterogeneity for E3SM applications.

L424: Our results demonstrate the accuracy and efficiency advantages of global unstructured meshes compared to a single structured grid.

A3. One of the strengths of using the unstructured approach in global simulations is highlighted nicely by the sentence on L413-414. Could the authors elaborate on why 4000 m was chosen for the refinement? In this wave model setup, the largest wave length resolved is around 1200 m, and depth effects are not likely to become important until around 600 m or shallower. I imagine the choice has to do with ensuring smooth transitions but it would be nice to know more details.

The 4000m criteria was chosen so that the continental shelf break would be well-resolved as wave energy propagates toward the coast. In many regions, the transition between the deep and shallow water regimes occurs rapidly due to the steepness of the shelf break. Therefore, we wanted to select a value that would put the enhanced mesh resolution out past this point, while

not resolving the deep ocean. We found that 4km accomplished this nicely for the U.S. refinement region considered here. We have added an additional sentence on L139 to clarify:

Since depth effects on waves become important on the continental shelf, the 4000m value allows this transition region to be resolved, while not introducing extra resolution in the deep ocean.

A4. In the comparisons with buoy data, there seems to be a negative bias at many of the stations when using the unstructured mesh model. I can see why the 2 degree structured mesh model might have a positive bias but the former case is not so clear to me. This may be more of an open question but do the authors have an idea of why this might be occurring (e.g., differences in numerical scheme)? Is this just a feature for this particular setup or does the negative bias persist for different resolution refinements (e.g., unstructured mesh with a 2 degree to a 1 degree transition)? As a potential follow up, what is the beta parameter value used for the ST4 physics? In Rascle and Ardhuin (Ocean Modelling, 2013), the beta value is tuned for use with NCEP CFSR winds using a structured mesh setup. It seems plausible that similar improvements could be made tuning the same parameter with an unstructured mesh setup.

We believe the consistent under-prediction is most likely due to differences in the numerical schemes, as you have suggested. Since it is a consistent bias, we agree that the beta value could be a useful means of correcting this. For this study, we used the model's default values for the ST4 source terms. This corresponds to the TEST 471 configuration from the WAVEWATCH III User's Manual (WAVEWATCH III® Development Group, 2019) with a  $\beta_{max}$  of 1.43. This is mid-way between the values optimized for the CRSR and ECMWF winds in Rascle and Ardhuin (Ocean Modelling, 2013). While each resolution could be tuned for optimized performance, we felt that keeping the parameters consistent through the set-ups would highlight the impact of the resolution and grid choices. We acknowledge that further improvement could likely be made for each grid set-up by further tuning. We have added the following paragraph in the discussion to highlight this:

As noted, the unstructured mesh results tend to consistently under-predict observations in swell-dominated regions. This is likely due to the different numerical scheme which may be more diffusive. Since the bias is consistent, tuning the  $\beta_{max}$  value as was done in Rascle and Ardhuin (2013) may be a viable way to reduce this bias. However, in this study, we have chosen to keep all the parameters consistent across the mesh configurations.

We also added this sentence to L176 to make our parameter choice explicit:

The default model ST4 values are used with a  $\beta_{\text{max}}$  value of 1.43 This corresponds to the TEST471 values from the WW3 user manual (WAVEWATCH III® Development Group, 2019).

A5. L418-420 & Table 3: Can you clarify the following statements in L418-420? "Overall, the ... have larger errors for the unstructured case." Is this referring to the buoy validation results or global comparisons with the 0.5 degree structured mesh? In the latter case, it does appear that the biggest improvements in agreement coincide with the Gulf of Mexico and the U.S. East Coast. However, in Table 3, there are dramatic performance increases in coastal regions of the Gulf of Maine, Alaska, and S. CA Coast when using the unstructured WW3. Aside from the N. CA and Pacific N. W. Coast, this improvement tends to be in regions with significant swell.

Thank you for this comment. We have edited the end of the discussion section to make this point clear:

Overall, when compared with the 1/2 degrees structured mesh, the unstructured mesh seems to perform very well in coastal wind-sea dominated environments, such as the Gulf of Mexico and U.S. East Coast. Swell-dominated environments, such as the U.S. West Coast and Alaska, tend to have larger errors for the unstructured case, but still represent an overall improvement in accuracy compared to the 2 degree structured mesh.

**Specific comments:**

**B1. L188-189: How much does the use of the PR1 propagation switch affect accuracy for the structured meshes? This seems like important information to include since most structured mesh setups would not likely use this setting.**

We agree this is an important point to address. We have include Figure 1 to show that while the PR1 switch is indeed less accurate than the PR3+UQ switches used for standard structured mesh configurations, the minor differences would not alter the main conclusions of our study. We felt that use of the first-order switch for the structured meshes was important to make fair comparisons between the structured and unstructured mesh, which also uses a first-order method, both in terms of accuracy and computational performance. We have added Figure 1 in an Appendix to the manuscript.

**B2. Section 4.1: What model values are being used for the buoy station comparisons? Is a nearest neighbor approach being used (i.e., using the cell average computed by WW3 that the station falls within)? And if not, how sensitive are the results to interpolation method?**

For the station comparisons, the interpolation is linear for the unstructured mesh and bilinear for the structured meshes. Since the numerical scheme used for the unstructured mesh is based on linear basis functions, linear interpolation results in the actual numeric solution at any point within an element. The situation is less clear for the structured mesh solution, since a flux reconstruction would be necessary to formally evaluate the numerical solution. However, in this case, bilinear interpolation is the standard approach that WW3 uses to compute station time series output. Therefore, we have not assessed the sensitivity of our results to other interpolation methods.

**B3. L265-266: Is buoy station 44025 relatively far from the coast compared with other the buoy stations?**

It is true that station 44025 is fairly close to the coast. We have removed "which are shallow stations further from the coast". The corrected text in L265-266 now reads:

The exceptions in this range are stations 44008 and 44025, which are shallow stations where all models provide similar accuracy.

**B4. L269: I recommend changing "The representation" to "Better representation", or similar.**

Thank you for the suggestion. This sentence in the paper does lack clarity. It was intended to point out that for two similarly placed stations on the shelf break, the unstructured mesh performs differently. This is likely due to the unstructured mesh representing the shelf-break more accurately at one station vs. the other. This sentence now reads:

The fidelity of the shelf break representation likely plays a role in the accuracy of the unstructured mesh solution, as exemplified by stations 44014 and 41025. The unstructured mesh provides the most accurate result at 44014, but does not improve as drastically over the 2 degree solution at a similarly placed shelf-break station to the south, at station 41025. This could occur because the mesh happens to more accurately represent the shelf-break near station 44014 compared to 41025.

**B5.** Table 3: It's a little odd that the 2 degree structured WW3 model performs slightly better in deep water than the 0.5 degree model for the S. CA Coast. Do the authors have any ideas why this might be?**

From examination of Figure 7 a) (reproduced here as Figure 2), the RMSE for the 2 degree solution is very slightly lower than the 1/2 degree model. We agree this difference seems to be amplified in the average errors in Table 3. We have provided Figure 3 here, which shows that indeed the relative bias of the 2 degree solution is slightly lower for the deep stations in this region (specifically 46047 and 46069). This indicates that the 2 degree mesh has slightly improved errors at the extremes. However, we contend that this improvement is in the noise and is not significant relative to the overall trends observed in the results.

**Minor comments:**

**C1. I believe the proper nomenclature for the model (in the title and manuscript) should be 'WAVEWATCH III'.**

Thank you for noticing this. We have added a space in the title and abstract. After the abstract, we use the abbreviation WW3 similar to other WAVEWATCH III publications.

**C2. L151-152: Please add 'degree' to both instances of "1/2 structured mesh/es".**

We have made this correction as suggested.

**C3. L295: Station should be '46215' and not '36215'.**

Thank you for catching this error; we have corrected it.

**C4. Table 3: Please state that table values are percentages.**

We have indicated the Table 3 values are percentages as suggested.

**Interactive comment:**

**The unstructured mesh presented here refines U.S. coastal regions at around 0.5 degrees using around 16,000 nodes. Does anyone know approximately how many nodes would be necessary to refine coastal regions for the entire globe?**

This is an interesting point; we have generated a mesh to estimate this as shown in Figure 4. It uses the 4000m depth criteria within 1000km of the coast for the entire globe. This mesh contained 80,423 nodes, which is about 5 times the number of nodes in the unstructured mesh for the U.S. used in this study.

**References**

- Rascle, N. and Ardhuin, F.: A global wave parameter database for geophysical applications. Part 2: Model validation with improved source term parameterization, Ocean Modelling, 70, 174–188, 2013.
- WAVEWATCH III® Development Group: User manual and system documentation of WAVEWATCH III® version 6.07, Tech. Note 333, NOAA/NWS/NCEP/MMAB, College Park, MD, USA, https://github.com/NOAA-EMC/WW3, 465 pp. + Appendices, 2019.

---

## Author Comment (AC3) · 10 Mar 2021

Please find our author response for this comment included at the end of the AC2 supplement: https://gmd.copernicus.org/preprints/gmd-2020-351/gmd-2020-351-AC2-supplement.pdf

---

## Author Response (AR1)

**Response to Reviewer #1**

**The manuscript is well-written and the authors set out their objectives and results very clearly. It is suitable for publication in GMD if the following comments are addressed.**

Thank you for reading through our manuscript and for providing valuable comments. Please find our point-by-point responses to your comments below. Reviewer comments are in bold and author responses are in plain text. All line numbers refer to the track-changes version of the manuscript provided in this document following the author responses.

**Specific comments:**

**Line 25: why are phase-averaged wave models quite expensive compared to the atmospheric and oceanic dynamic models? Do you have evidences for this? By the way, are you talking about "spectral wave models" or "phase-averaged" (which is more general,e.g. XBeach).**

**Author Response:**

The expense of the wave model is due to the need to evolve the frequency/direction spectrum of the wave action in longitude/latitude space. This leads to a greater number of overall degrees of freedom compared to an ocean or atmospheric models. In the case of our study, the spectral mesh resolution means that there are 1800 unknowns per grid point (50 frequencies  $\times$ 36 directions), as compared to a typical low-resolution E3SM configuration where the ocean and atmosphere models have less than 100 vertical layers at each grid point. The low-resolution E3SMv1 model ran at 10 simulated years per day (Golaz et al., 2019), which is roughly what we found the throughput of the unstructured mesh solution to be in this study. For additional evidence, we also point to the study done in Li et al. (2016) where WAVEWATCH III coupled to the CESM model. In order to reduce cost relative to the rest of the coupled model, the wave model resolution used in that paper was very coarse:  $3.2 \times 4$ degree with a 25 frequency and 24 direction spectral grid.

Also, in the submitted manuscript we did use "spectral wave model" and "phase-averaged" wave model interchangeably. Based on your comment, we now realize that it is more accurate to refer to spectral wave models as a specific class of phase-averaged model. We have done our best to clear this up in the text by referring to WAVEWATCH III as a "phase-averaged, spectral wave model"

**Changes in Manuscript:**

Line 25: However, even phase-averaged, spectral wave models are quite expensive compared to the atmospheric and oceanic dynamic cores used in Earth system models.

Line 27: Since phase-averaged spectral wave models are known to be expensive, variable resolution approaches can be used to economically resolve both the coastal and global regimes.

Line 69: These phase-averaged spectral wave models describe the evolution of the wave action density spectrum,  $N(\lambda, \phi, k, \theta, t)$ , which is a function of both longitude/latitude  $(\lambda, \phi)$  and wavenumber/direction  $(k, \theta)$  space.

Caption for Figure 1: Energy spectrum for wave frequencies in the ocean showing the role of phase-averaged, spectral wave models.

**Line 113: please add reference Dietrich et al. (2011) for completeness.**

Author Response:

We have added this reference, as suggested.

Changes in Manuscript:

Line 117: To date, unstructured meshes have been primarily used in regional studies to assess accuracy in coastal settings (Roland and Ardhuin, 2014; Abdolali et al., 2020) (Roland and Ardhuin, 2014; Abdolali et al., 2020; Dietrich et al., 2011).

**Line 188: the first order PR1/CRD-N schemes have been employed. To what extent do they influence the outcomes (i.e. less accurate) in particular the 2-degree mesh case?**

**Author Response:**

Thank you for raising this point. We have done our best to address this in Figure A1 (Reproduced here as Figure R1). This figure shows that the standard PR3+UQ schemes for structured meshes are more accurate than the first order PR1 switch. However, the differences do not alter the conclusions of our study. We have added this figure as an Appendix in the manuscript starting on Line 460.

**Changes in Manuscript:**

**Appendix A:** Differences between first- and third-order schemes for structured meshes As mentioned in Section 3.4, typical WW3 structured mesh configurations use the third-order ULTIMATE QUICKEST scheme instead of the first-order method used here for comparison with the first-order unstructured CRD-N scheme. Figure A1 shows the accuracy differences between the PR1 and PR3/UQ switches for the 1/2 degree and 2 degree structured meshes. Each subplot represents validation results from buoys in the regions from Sections 3-10. These plots demonstrate that while the PR3/UQ scheme is slightly more accurate overall, the choice of numerical scheme for the structured meshes does not change the conclusions of our study.

**Figure R1.** Root mean square errors comparing the accuracy differences between the PR1 and PR3+UQ switches. The 2 degree structured mesh is represented by orange and the 1/2 structured mesh is shown in purple. Darker shades are associated with the PR3 configuration while lighter shades correspond to the PR1 switch. Subplots are presented in order of regions from the manuscript as follows: (a) Gulf of Main, (b) South to Mid-Atlantic East Coast, (c) Gulf of Mexico, (d) Caribbean Region, (e) Southern California Coast, (f) Northern California and Pacific Northwest Coast, (g) Alaskan Coast, (h) Hawaiian Coast.

**Technical corrections :**

**Line 295: 36215 -> 46215**

Author Response: Thank you for carefully reading through the text to catch this mistake.

Changes in Manuscript:

 $\overline{\text{Line 306: At station } \frac{3621546215}{3621546215}}$ , for example, the unstructured mesh is the least accurate solution.

**Response to Reviewer #2**

This is a groundbreaking and well-written study. The authors demonstrate how an unstructured WAVEWATCH III (WW3) spectral wave model setup can be used to conduct variable-resolution global simulations, with purposes of coupling with an earth system model. Like the two-way nested structured mesh approach, the unstructured model resolves coastal areas with similar accuracy of a higher resolution uniform-structured-mesh model but with lower computational costs. The study uses a consistent method to determine global mesh refinement. The unstructured approach is straight-forward to integrate with earth system models, and is an ideal choice for coupling with variable-resolution earth system models such as the E3SM.

Thank you for reading through our manuscript and for providing thoughtful comments. They were very useful in helping us improve the message of this paper. Please find our point-by-point responses to your comments below. Reviewer comments are in bold and author responses are in plain text. All line numbers refer to the track-changes version of the manuscript provided in this document following the author responses.

**General comments:**

A1. The first half of Section 2.1 (L91-98) seems to imply that using a two-way nested structured mesh setup in WW3 is more computationally expensive than a variable resolution unstructured mesh (in terms of achieving similar accuracy). Is this a correct interpretation of the text? And if so, is it more of an opinion or is it based on previous studies or work by the authors? I ask because regional wave climate studies often use WW3 with a nested structured mesh setup to resolve coastal regions and readers might be curious how this approach compares with the unstructured mesh setup. My understanding is that there is a higher computational overhead (or at least was initially) when using an unstructured mesh (as well as differences in accuracy). Depending on the setup, it seems highly plausible that a two-way nested model could be more accurate or faster than using the unstructured approach (despite calculating potentially overlapping cells). While a comparison of these two approaches is likely outside the scope of this study, it seems like a potentially missing component to make any definitive statements on accuracy or cost.

**Author Response:**

That is the correct interpretation of the text, but, as of now, it is based more on educated conjectures than previous studies or recent work by the authors. We agree that more direct comparisons with nested grids and SMC are out of scope of this current study, but would be required for definitive statements on accuracy or cost. To not imply direct comparisons and better reflect our scope, Section 2.1 was rewritten as follows:

**Changes in Manuscript:**

Lines 93-106: The two-way nested mosaic approach has been extensively validated against historical wave observations and is used for NOAA forecasting operations (Chawla et al., 2013a). However, two-way nesting of structured meshes has several disadvantages (Zijlema, 2010). In these types of meshes, transitions in resolution are typically abruptand must therefore. Therefore, they must either be placed well outside regions where high resolution is needed in order to be accurate or a series of nested meshes must be employed to achieve a smooth transition. This means high resolution regions must be larger than necessary to avoid degrading accuracy, which incurs additional computational cost. In addition, or increasing the complexity of the nested model. Two-way nesting also requires a sufficient overlap regionis required to accomplish the two-way nesting, which means duplicate calculations are performed in these regions on both the coarse and fine meshes. Abrupt transitions in resolution can be mitigated by increasing the number of nested grids, at the expense of complexity. SMC grids provide an alternative multi-resolution capability. However, similar to nested meshes, SMC meshes also lack the ability to smoothly vary resolution in a flexible manner. Another option is to nest an unstructured coastal wave model, such as SWAN (Zijlema, 2010), inside a global WW3 domain (Amrutha et al., 2016). However, this approach uses the WW3 wave spectrum solution to force the boundary of the nested SWAN model, which only provides a one-way coupling between the models. For coupled E3SM applications, two-way feedbacks between the coastal to the global ocean within the wave model are desired.

A2. As a follow up, I am wondering if this manuscript has overlooked situations where a nested structured mesh setup with WW3 might be preferable to an unstructured mesh for global simulations. Of course, it seems quite natural and desirable to use WW3 with an unstructured mesh with variable resolution earth system models such as the E3SM. However, there are some earth system models that do not use variable resolutions (e.g., ocean circulation model with a fixed horizontal spatial resolution) but do employ high-resolution atmospheric models (e.g., 0.2 degree spectral grid). In such situations, it seems like the nested structured mesh approach might be better since communication with the atmospheric component would require limited field remapping (assuming near-coincident mesh placement). In light of this, it might be worthwhile to review a few statements, such as those found in L47, L109, and L424-426, to see if any minor clarifications are necessary.

**Author Response:**

We agree and acknowledge that the multi-grid nested case and the SMC grid case were not considered in this study. We limited the investigation to structured meshes and unstructured triangle meshes. Minor clarifications were made to better express this.

**Changes in Manuscript:**

Lines 51-53: The purpose of this paper is to report on progress toward this goal, starting with an assessment of the accuracy and performance of the WW3 model (Tolman, 1991) using global to coastal unstructured meshes, which are compared to a single structured grid.

Lines 112-113: However, they are less applicable to global Earth system modeling due to not considered here due to their added complexity and heterogeneity for E3SM applications.

Lines 440-441: Our results demonstrate the accuracy and efficiency advantages of global unstructured meshes compared to a single structured grid

A3. One of the strengths of using the unstructured approach in global simulations is highlighted nicely by the sentence on L413-414. Could the authors elaborate on why 4000 m was chosen for the refinement? In this wave model setup, the largest wave length resolved is around 1200 m, and depth effects are not likely to become important until around 600 m or shallower. I imagine the choice has to do with ensuring smooth transitions but it would be nice to know more details.

**Author Response:**

The 4000m criteria was chosen so that the continental shelf break would be well-resolved as wave energy propagates toward the coast. In many regions, the transition between the deep and shallow water regimes occurs rapidly due to the steepness of the shelf break. Therefore, we wanted to select a value that would put the enhanced mesh resolution out past this point, while not resolving the deep ocean. We found that 4km accomplished this nicely for the U.S. refinement region considered here. We have added an additional sentence to clarify.

**Changes in Manuscript:**

Line 144-146: Since depth effects on waves become important on the continental shelf, the 4000m value allows this transition region to be resolved, while not introducing extra resolution in the deep ocean.

A4. In the comparisons with buoy data, there seems to be a negative bias at many of the stations when using the unstructured mesh model. I can see why the 2 degree structured mesh model might have a positive bias but the former case is not so clear to me. This may be more of an open question but do the authors have an idea of why this might be occurring (e.g., differences in numerical scheme)? Is this just a feature for this particular setup or does the negative bias persist for different resolution refinements (e.g., unstructured mesh with a 2 degree to a 1 degree transition)? As a potential follow up, what is the beta parameter value used for the ST4 physics? In Rascle and Ardhuin (Ocean Modelling, 2013), the beta value is tuned for use with NCEP CFSR winds using a structured mesh setup. It seems plausible that similar improvements could be made tuning the same parameter with an unstructured mesh setup.

**Author Response:**

We believe the consistent under-prediction is most likely due to differences in the numerical schemes, as you have suggested. Since it is a consistent bias, we agree that the beta value could be a useful means of correcting this. For this study, we used

the model's default values for the ST4 source terms. This corresponds to the TEST 471 configuration from the WAVEWATCH III User's Manual (WAVEWATCH III® Development Group, 2019) with a  $\beta_{max}$  of 1.43. This is mid-way between the values optimized for the CRSR and ECMWF winds in Rascle and Ardhuin (Ocean Modelling, 2013). While each resolution could be tuned for optimized performance, we felt that keeping the parameters consistent through the set-ups would highlight the impact of the resolution and grid choices. We acknowledge that further improvement could likely be made for each grid set-up by further tuning. We have added text in two places to highlight this:

**Changes in Manuscript:**

Lines 430-433: As noted, the unstructured mesh results tend to consistently under-predict observations in swell-dominated regions. This is likely due to the different numerical scheme which may be more diffusive. Since the bias is consistent, tuning the  $\beta_{\text{max}}$  value as was done in Rascle and Ardhuin (2013) may be a viable way to reduce this bias. However, in this study, we have chosen to keep all the parameters consistent across the mesh configurations.

Line 184: The default model ST4 values are used with a  $\beta_{max}$  value of 1.43 This corresponds to the TEST471 values from the WW3 user manual (WAVEWATCH III® Development Group, 2019).

A5. L418-420 & Table 3: Can you clarify the following statements in L418-420? "Overall, the ... have larger errors for the unstructured case." Is this referring to the buoy validation results or global comparisons with the 0.5 degree structured mesh? In the latter case, it does appear that the biggest improvements in agreement coincide with the Gulf of Mexico and the U.S. East Coast. However, in Table 3, there are dramatic performance increases in coastal regions of the Gulf of Maine, Alaska, and S. CA Coast when using the unstructured WW3. Aside from the N. CA and Pacific N. W. Coast, this improvement tends to be in regions with significant swell.

**Author Response:**

Thank you for this comment. We have edited the end of the discussion section to make this point clear:

**Changes in Manuscript:**

Lines 434-437: Overall, when compared with the 1/2 degrees structured mesh, the unstructured mesh seems to perform very well in coastal wind-sea dominated environments, such as the Gulf of Mexico and U.S. East Coast. Swell-dominated environments, such as the U.S. West Coast and Alaska, tend to have larger errors for the unstructured case, but still represent an overall improvement in accuracy compared to the 2 degree structured mesh.

**Specific comments:**

**B1.** L188-189: How much does the use of the PR1 propagation switch affect accuracy for the structured meshes? This seems like important information to include since most structured mesh setups would not likely use this setting.**

**Author Response:**

We agree this is an important point to address. We have include Figure A1 (reproduced here as Figure R2) to show that while the PR1 switch is indeed less accurate than the PR3+UQ switches used for standard structured mesh configurations, the minor differences would not alter the main conclusions of our study. We felt that use of the first-order switch for the structured meshes was important to make fair comparisons between the structured and unstructured mesh, which also uses a first-order method, both in terms of accuracy and computational performance. We have added Figure A1 in an Appendix to the manuscript starting on Line 460.

**Changes in Manuscript:**

**Appendix A:** Differences between first- and third-order schemes for structured meshes As mentioned in Section 3.4, typical WW3 structured mesh configurations use the third-order ULTIMATE QUICKEST scheme instead of the first-order method used here for comparison with the first-order unstructured CRD-N scheme. Figure A1 shows the accuracy differences between the PR1 and PR3/UQ switches for the 1/2 degree and 2 degree structured meshes. Each subplot represents validation results from buoys in the regions from Sections 3-10. These plots demonstrate that while the PR3/UQ scheme is slightly more accurate overall, the choice of numerical scheme for the structured meshes does not change the conclusions of our study.